# Biochemical and functional characterization of a meiosis-specific Pch2/ORC AAA+ assembly

María Ascensión Villar-Fernández[1,2], Richard Cardoso da Silva[1] , Magdalena Firlej[3], Dongqing Pan[1] , Elisabeth Weir[1], Annika Sarembe[1], Vivek B Raina[1,2] , Tanja Bange[1], John R Weir[1,3] , Gerben Vader[1]

**Pch2 is a meiosis-specific AAA+ protein that controls several important chromosomal processes. We previously demonstrated that Orc1, a subunit of the ORC, functionally interacts with budding yeast Pch2. The ORC (Orc1-6) AAA+ complex loads the AAA+ MCM helicase to origins of replication, but whether and how ORC collaborates with Pch2 remains unclear. Here, we show that a Pch2 hexamer directly associates with ORC during the meiotic G2/prophase. Biochemical analysis suggests that Pch2 uses its non-enzymatic NH$_2$-terminal domain and AAA+ core and likely engages the interface of ORC that also binds to Cdc6, a factor crucial for ORC-MCM binding. Canonical ORC function requires association with origins, but we show here that despite causing efficient removal of Orc1 from origins, nuclear depletion of Orc2 and Orc5 does not trigger Pch2/Orc1-like meiotic phenotypes. This suggests that the function for Orc1/Pch2 in meiosis can be executed without efficient association of ORC with origins of replication. In conclusion, we uncover distinct functionalities for Orc1/ORC that drive the establishment of a non-canonical, meiosis-specific AAA+ assembly with Pch2.**

## Introduction

Meiosis is a specialized cell division program that produces haploid gametes that are required for sexual reproduction. Ploidy reduction requires several meiosis-specific events, which occur in the context of a highly orchestrated meiotic program (Petronczki et al, 2003). During the meiotic G2/prophase, Spo11-dependent DNA double-strand breaks (DSBs) are repaired via homologous recombination (Lam & Keeney, 2015). DSB formation and recombination are essential for meiosis, but errors that occur during these events endanger genome stability of the developing gametes (Sasaki et al, 2010).

Pch2 (known as TRIP13 in mammals) is an AAA+ protein that controls multiple aspects of meiotic homologous recombination and checkpoint signaling (San-Segundo & Roeder, 1999; Bhalla & Dernburg, 2005; Li & Schimenti, 2007; Borner et al, 2008; Joshi et al, 2009; Roig et al, 2010; Vader et al, 2011; Vader, 2015). AAA+ proteins are ATPases that, via cycles of nucleotide binding and hydrolysis, can undergo conformational changes to influence a wide range of client molecules (reviewed in Hanson and Whiteheart [2005]). A characteristic of AAA+ proteins is their ability to assemble into ring-shaped homo- or hetero-hexamers, often mediated through interactions between AAA+ domains. Pch2 forms homo-hexamers and uses its enzymatic activity to remodel (and affect the function of clients) (Chen et al, 2014; Ye et al, 2015; Ye et al, 2017; Alfieri et al, 2018). HORMA domain–containing proteins are confirmed Pch2 clients, and many, if not all, functions ascribed to Pch2 can be explained by its enzymatic activity toward HORMA proteins (Vader, 2015).

Because recruitment of Pch2 to chromosomes is associated with (at least some) Pch2 functions, it is imperative to understand how Pch2 is recruited to meiotic chromosomes and whether adaptor proteins are required to facilitate specific functions of this AAA+ protein. In addition to its global role in controlling meiotic recombination, Pch2 is needed to prevent inappropriate DSB formation and recombination within the repetitive ribosomal DNA (rDNA) array of budding yeast (San-Segundo & Roeder, 1999; Vader et al, 2011). In line with a role for Pch2 in promoting rDNA stability, Pch2 is enriched within the nucleolus, the nuclear compartment where the rDNA resides. This rDNA-specific recruitment and function of Pch2 requires Orc1 (Vader et al, 2011), raising the interesting possibility that Orc1 could fulfill a meiosis-specific role by interacting with a meiosis-specific AAA+ protein complex.

Orc1 is a component of the origin recognition complex (ORC), a central mediator of several key chromosomal processes. The ORC is a hetero-hexameric protein complex composed of Orc1 through Orc6, wherein Orc1-5 are AAA+ proteins, and Orc6 shows no structural similarity with the rest of ORC components (reviewed in

[1]Department of Mechanistic Cell Biology, Max Planck Institute of Molecular Physiology, Dortmund, Germany   [2]International Max Planck Research School in Chemical and Molecular Biology, Max Planck Institute of Molecular Physiology, Dortmund, Germany   [3]Friedrich Miescher Laboratory, Tübingen, Germany

Correspondence: gerben.vader@mpi-dortmund.mpg.de
Dongqing Pan's present address is Department of Structural Biology, Graduate School of Pharmaceutical Sciences, Kyoto University, Kyoto, Japan
Elisabeth Weir's present address is Max Planck Institute for Developmental Biology, Tübingen, Germany
Annika Sarembe's present address is Institut für Pathologie am Klinikum Leverkusen, Leverkusen, Germany
Tanja Bange's present address is Institute of Medical Psychology, LMU Munich, Munich, Germany

Bell and Kaguni [2013] and Bell and Labib [2016]). A key role for the ORC in the chromosome metabolism lies in its function in the initiation of the eukaryotic DNA replication. The ORC binds to origins of replication, which in budding yeast are defined by a specific DNA sequence. Such sequence specificity seems to be absent in *Schizosaccharomyces pombe* and metazoans, in which origins of replication are predominantly determined by the chromatin structure, epigenetic marks, and specifically, the presence of a nucleosome-free region (Peng et al, 2015; Prioleau & MacAlpine, 2016). At origins, interaction of Orc1-6 with Cdc6, another AAA+ protein, through typical AAA+-to-AAA+ interactions, creates a hexameric ORC-Cdc6 assembly. ORC-Cdc6 (with the help of additional proteins) drives the localized, chromosomal recruitment of the AAA+ MCM helicase (Bell & Labib, 2016). The MCM helicase forms the basis of the replication complex and as such is key to initiate DNA replication. This canonical MCM loader function of the ORC occurs at origins of replication during the G1 phase of the cell cycle and is essential for DNA replication.

In addition to the role of the ORC in DNA replication, the ORC executes several other chromosomal functions during the mitotic cell cycle. In budding yeast, the ORC is required for transcriptional gene silencing at the cryptic mating-type loci and at telomeres (Foss et al, 1993; Loo et al, 1995; Fox et al, 1997). At the mating loci, the ORC recruits Sir1, a silencing factor, via a direct association between Sir1 and Orc1 (Triolo & Sternglanz, 1996; Hou et al, 2005). Furthermore, the budding yeast ORC is also involved in regulating sister chromatid cohesion (Suter et al, 2004; Shimada & Gasser, 2007). Importantly, all these roles of the ORC are associated with binding of the ORC to origins of replication, which has led to a model wherein the ORC functions as a structural "loading platform" for specific interacting proteins which subsequently allows for the establishment of localized chromosomal activities at origins of replication (Bell & Labib, 2016).

Here, we use in vivo analysis during budding yeast meiosis, coupled to in vitro biochemical reconstitution to investigate how Pch2 interacts with Orc1/ORC. We find that Pch2 directly engages the entire ORC in the meiotic G2/prophase, in a manner that is consistent with an AAA+ to client/adaptor relationship. Surprisingly, we find that depletion of Orc2 or Orc5 does not trigger phenotypes that are associated with Pch2 function at rDNA. These findings are in contrast to earlier depletion studies, using hypomorphic alleles of Orc1, which revealed a key contribution for Orc1 to Pch2 function (Vader et al, 2011). Our in vitro analysis provides biochemical insights into the interaction between Pch2 and ORC. Chemical cross-linking combined with mass spectrometry (XL-MS), coupled to biochemical characterization, shows that the ORC–Pch2 interaction is distinct from the well-established interaction between ORC and MCM. In contrast to the ORC–MCM assembly, the ORC–Pch2 assembly does not require Cdc6. Our data suggest that Orc1 plays a key role in the interaction between ORC and Pch2, which is in agreement with the key role of Orc1 in mediating the function of Pch2 in maintaining rDNA stability. As a whole, our data point to the existence of a distinct function for Orc1/ORC during meiosis, during which it interacts with Pch2.

## Results

Pch2 functionally interacts with Orc1 to protect rDNA border regions from instability during meiosis (Vader et al, 2011), and we aimed to

further understand the interaction between these two AAA+ proteins. We first investigated how this interaction depended on Pch2 hexamer formation and ATP hydrolysis activity in vivo. We used an ATP hydrolysis mutant within the Walker B domain of Pch2 (*pch2–E399Q*) (Fig 1A), which is unable to support rDNA-associated DSB protection (Vader et al, 2011). In other AAA+ enzymes, mutating this critical residue in the Walker B domain prevents efficient ATP hydrolysis and stalls the stereotypical catalytic cycle of AAA+ enzymes. This often leads to stabilized interactions between AAA+ proteins and their clients and/or adaptors. Equivalent mutants in other AAA+ enzymes have been used to trap enzyme–client and/or enzyme–adaptor interactions (Hanson & Whiteheart, 2005; Ritz et al, 2011). We detected an increased interaction between Pch2 and Orc1 in meiotic cells expressing Pch2-E399Q as compared with cells expressing wild-type Pch2 (Figs 1B and S1A). We also detected a robust interaction between Orc1 and Pch2 in lysates where DNA was degraded (Fig S1B and C), excluding potential indirect association mediated by DNA present in our co-immunoprecipitation assays. We next investigated a different mutant Pch2 allele, which carried a mutation within the Walker A motif (K320R). Mutations in residues located within this motif have been shown to reduce ATP binding (Hanson & Whiteheart, 2005). When we probed the interaction between Pch2 and Orc1, Orc1–TAP failed to co-immunoprecipitate Pch2–K320R (Fig 1A and C). Considering that mutations in the Walker A motif lead to monomerization of Pch2 in vivo (Herruzo et al, 2016), our data suggest that the efficient interaction between Pch2 and Orc1 relies on ATP binding and Pch2 hexamer formation. As a whole, these experiments indicate that Pch2 interacts with Orc1 in a manner that is consistent with a stereotypical AAA+/client and/or adaptor interaction.

Many, if not all, functions ascribed to Orc1 involve its assembly into the six-component ORC (consisting of Orc1-6) (Bell & Labib, 2016). We therefore tested whether in addition to Orc1, other subunits of the ORC also interacted with Pch2. We used the *pch2-E399Q* allele to stabilize in vivo interactions. Our co-immunoprecipitation (co-IP) assays revealed that TAP-tagged versions of Orc2/Orc5 (Fig S1A) co-immunoprecipitated with Pch2 during the meiotic G2/prophase (Fig 1D and E). Similarly, we could pulldown 3xFlag-tagged Pch2–E399Q with Orc2 using an α-Orc2 antibody (Fig 1F). Furthermore, an unbiased mass spectrometric analysis of the Pch2–E399Q interactome identified Orc5 in addition to Orc1, indicating that Pch2 interacts with multiple ORC subunits (VB Raina and G Vader, unpublished observations). Altogether, we conclude that in addition to Orc1, Pch2 can in vivo associate with other subunits of the ORC during the meiotic G2/prophase. We consistently observed a strong association between Pch2 and Orc1 relative to the other subunits tested in our comparative in vivo co-IP experiments (Fig 1E), which together with other data (XL-MS, co-IP assays, pulldowns and functional analysis; see below) suggest that Orc1 might be a central interactor of Pch2.

Our in vivo analysis demonstrated that Pch2 associates with Orc1/ORC, and we sought to test whether this association is direct. For this, we expressed and purified budding yeast Pch2 (carrying an NH$_2$-terminal His–MBP tag) through a baculovirus-based protein expression system. As judged by size exclusion chromatography (SEC), purified Pch2 assembled into an apparent hexamer (predicted size ~636 kD), with a minor fraction that appears to be monomeric (size of ~106 kD for His–MBP–Pch2) (Fig 2A). We confirmed the

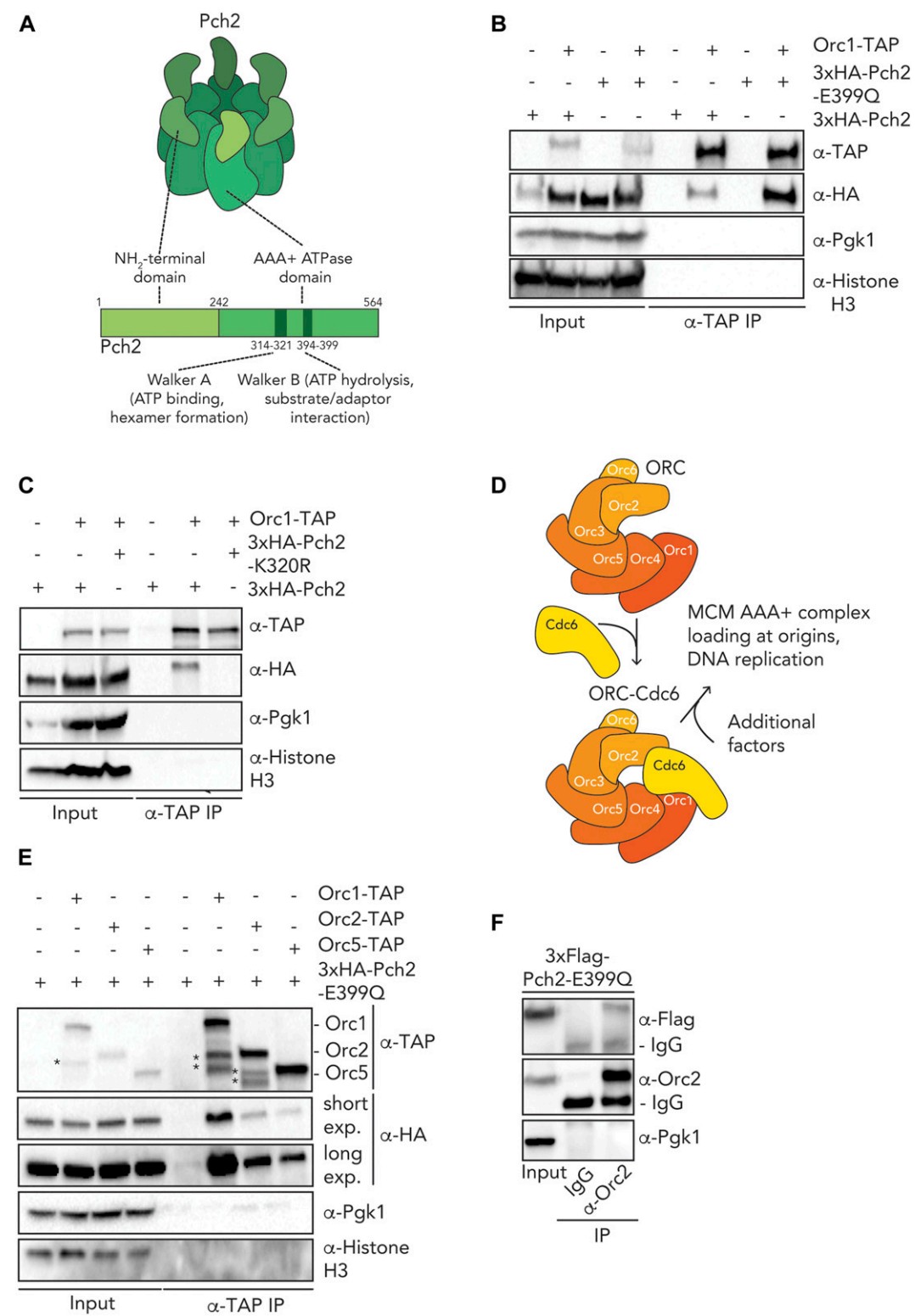

**Figure 1. In vivo characterization of ORC-Pch2.**
**(A)** Schematic of hexameric Pch2 AAA+ assembly, with domains organization of Pch2. **(B)** Co-immunoprecipitation (co-IP) of wild-type Pch2 and Pch2–E399Q with Orc1–TAP (via α-TAP-IP) during the meiotic prophase (5 h into meiotic program). **(C)** co-IP of wild-type Pch2 and Pch2–K320R with Orc1–TAP (via α-TAP-IP) during the meiotic prophase (5 h into meiotic program). **(D)** Schematic of the Orc1-6/AAA+ complex and its canonical role with the Cdc6 AAA+ protein (and additional factors) in MCM/AAA+ complex loading and DNA replication. **(E)** co-IP of Pch2–E399Q with Orc1–TAP, Orc2–TAP, and Orc5–TAP during the meiotic prophase (5 h into meiotic program) (via α-TAP-IP). For α-HA, short and long exposures are shown. * indicate degradation products of either Orc1–TAP or Orc2–TAP. **(F)** co-IP of Pch2–E399Q with Orc2 (via α-Orc2 IP). Isotype IgG IP was used as negative control.

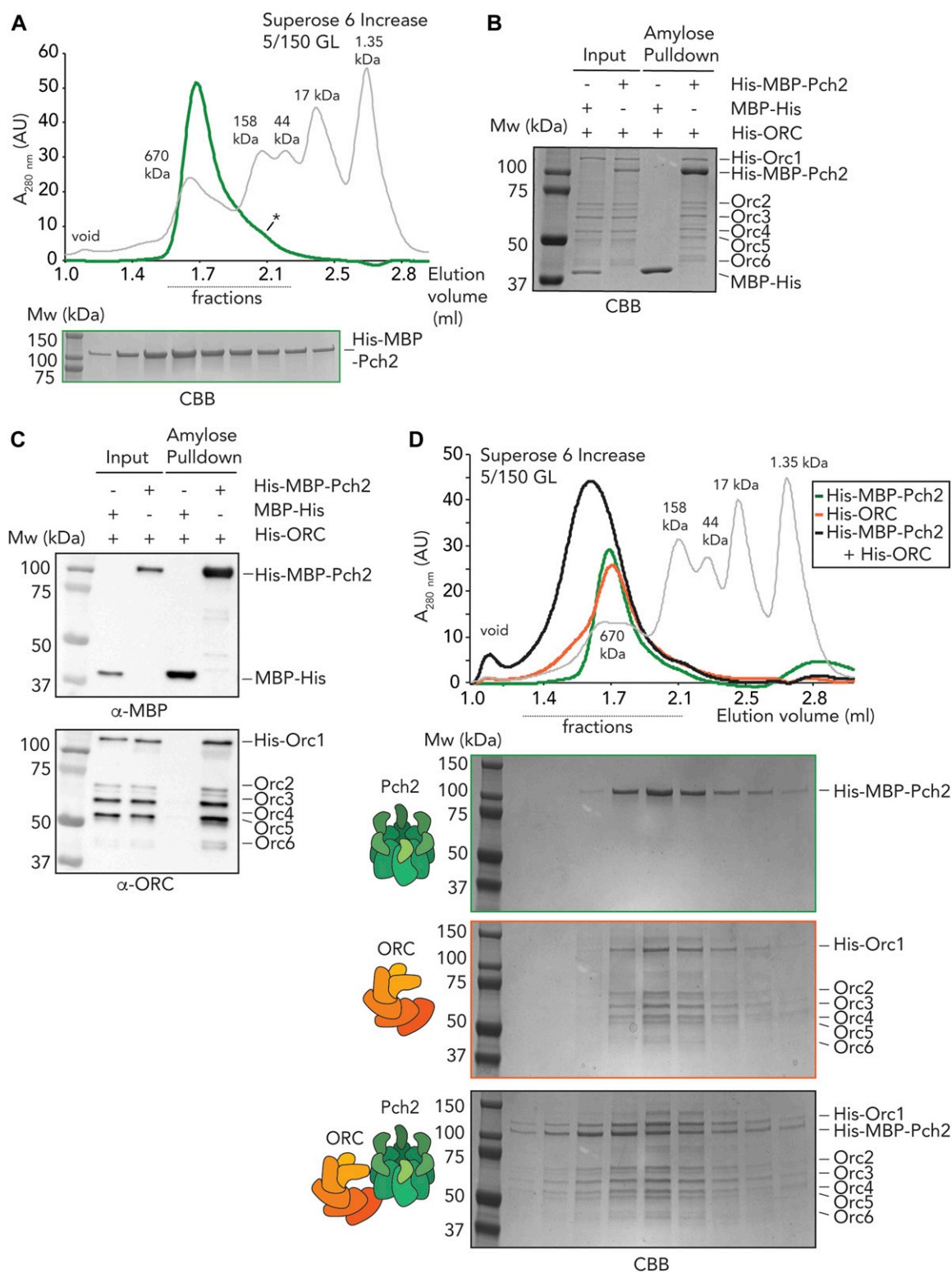

**Figure 2. In vitro reconstitution of the origin recognition complex (ORC)–Pch2 complex.**
**(A)** Size exclusion chromatography of His–MBP–Pch2 purified from insect cells. Coomassie Brilliant Blue (CBB) staining of peak fractions (dotted line) run on SDS–PAGE gel. * indicates likely monomeric fraction of His–MBP–Pch2. AU, arbitrary units. **(B, C)** Amylose-based pulldown of the ORC (His–Orc1-6 and His–ORC) purified from insect cells, with His–MBP–Pch2. **(B)** CBB staining, (C) Western blot analysis using α-MBP and α-ORC (which recognizes all six ORC subunits). **(D)** Size exclusion chromatography of His–ORC (His–MBP–Pch2) assembly. CBB staining of peak fractions (dotted line) run on SDS–PAGE gel. AU, arbitrary units.

functionality of our affinity-purified Pch2 by demonstrating a direct interaction with its substrate Hop1, as previously described (Chen et al, 2014) (Fig S2A). We note that performing the pulldown between Pch2 and Hop1 in the presence of ATP or a slow-hydrolysable form of ATP (ATP-γS) did not alter the binding behavior (Fig S2B), in contrast to what has been observed (Chen et al, 2014) and would be expected based on the AAA+ catalytic cycle. We do not currently know the basis of this difference, but we speculate that the purification of Pch2 from insect cells (as opposed to purification from budding yeast [Chen et al, 2014]) might yield a Pch2 hexamer that is not properly activated potentially because of lack of certain (yeast-specific) posttranslational modifications. Under such conditions, impairing ATP hydrolysis is not expected to influence interactions.

We next tested whether Pch2 directly interacted with the ORC. For this, we used either the ORC purified from insect cells, as described here (i.e., Orc1-6, with Orc1 carrying a His tag; total size ~414 kD) (see Fig S3A and B) or ORC (i.e., Orc1-6, with Orc1 carrying a CBP tag) purified from α-factor–arrested vegetative budding yeast cells, as described (e.g., see Yeeles et al [2015]) (see below). We confirmed the presence of all ORC subunits in the insect cell–purified ORC by mass spectrometry (MS) (Fig S3C), and the composition of the insect cell–purified ORC was comparable with the budding yeast–purified ORC (Fig S3D). We did observe that in the case of purification from insect cells, the Orc6 subunit of the ORC migrated as a double band (Fig S3D), suggesting that a fraction of Orc6 might be phosphorylated, as described (Nguyen et al, 2001; Weinreich et al, 2001). Indeed, treatment of the insect cell–purified ORC with λ-phosphatase caused a collapse of the double band into a single Orc6 band (Fig S3E).

Using solid-phase pulldown experiments, we found that Pch2 is able to interact with the entire ORC (i.e., Orc1-6), irrespective of the source of the ORC (i.e., insect cells [Fig 2B and C] or budding yeast [Fig S3G]). The dephosphorylated ORC also interacted with Pch2, showing that the phospho-status of Orc6 did not affect interaction with Pch2 (Fig S3F). These experiments demonstrate that these AAA+ proteins indeed interact directly. Next, we asked whether this interaction could also be reconstituted in solution. SEC analysis confirmed that the ORC and Pch2 form a complex in solution, as judged by a reduced retention volume (which is indicative of a larger and/or more elongated complex) when combined, as compared with the elution profiles of Pch2 or ORC individually (Fig 2D). We suggest that the ORC and Pch2 interact with each other in an ORC (Orc1-6 hexamer) to Pch2 (hexamer) fashion, yielding what would be a complex of ~1 MD. We finally tested the effect of nucleotides on binding between Pch2 and ORC. As we observed with regard to the interaction between Pch2 and Hop1, addition of ATP (or ADP, or nonhydrolysable analogs) did not affect binding behavior between Pch2 and ORC (Fig S3H). Because we did observe differences in the interaction between Pch2 and Orc1 in the meiotic G2/prophase (Fig 1B and C), we again speculate that these different behaviors can be attributed to differential Pch2 ATPase activity levels present in our in vitro preparations as compared with in vivo conditions. Taken together, these experiments demonstrate that Pch2 directly interacts with the ORC, and as such reveal a novel direct interaction partner of Pch2 during the meiotic G2/prophase.

We next focused on understanding how Pch2 associates with the ORC. Associations between AAA+ proteins typically rely on interdomain

AAA+ interactions (Hanson & Whiteheart, 2005). For example, interdomain AAA+ contacts between individual ORC subunits establish the hexameric ORC formation (Bell & Kaguni, 2013; Bell & Labib, 2016). Alternatively, many hexameric AAA+ ATPases (including TRIP13, the mammalian homolog of Pch2 [Alfieri et al, 2018; Ye et al, 2017]) associate with clients/adaptors via an initial engagement using their NTDs (as is the case for the interaction between Pch2/TRIP13 and HORMA domain proteins) and, subsequently, show interactions mediated through AAA+ core/client binding (Hanson & Whiteheart, 2005). We thus set out to investigate the contributions of the NTD and AAA+ domains of Pch2 to ORC association. We first tested whether the NTD of Pch2 was involved in Pch2–ORC assembly. For this, we used Y2H analysis to show that Pch2 lacking its NTD (amino acids 2–242) was unable to interact with Orc1 (Fig 3A and B). We next purified Pch2 lacking the NTD (His–MBP–Pch2-243-564) from insect cells. By SEC, we observed that this Pch2 protein eluted at an apparent size that indicated a more extended shape or less organized assembly as compared with full-length Pch2 (Fig S4A). These findings imply a role for the NTD in stabilizing and/or maintaining Pch2 into a stable, well-ordered hexamer (see also above). Importantly, the ability of purified Pch2-243-564 to interact with the ORC was abolished, further demonstrating an important contribution of the NTD of Pch2 in directing interaction with the ORC (Fig 3C). Finally, we investigated the interaction between Pch2 and Orc1 in the meiotic G2/prophase by expressing an identical truncated version of Pch2 (3xFlag-Pch2-243-564). This truncated version of Pch2 was impaired in its ability to interact with Orc1 (Fig 3D), similar to what was observed using Y2H and in vitro interaction studies, pointing to a crucial contribution of the NTD of Pch2 to establish binding with the ORC. The residual interaction of Pch2-243-564 with Orc1 in the meiotic G2/prophase might indicate that under physiological conditions, Pch2 lacking its NTD retains a certain degree of affinity toward ORC (Fig 3D). Pch2 protects rDNA array borders (i.e., the ~1–10 outermost rDNA repeats and ~50 kb of single-copy flanking sequences) against meiotic DSB formation ([Vader et al, 2011], and Fig 3E). In agreement with an important role of the NTD of Pch2 in mediating Pch2 function (and Orc1/ORC association) during the meiotic G2/prophase, we found that cells expressing 3xFlag-Pch2-243-564 (Fig 3F) exhibited rDNA border–associated DSB formation, as is observed in pch2Δ cells (Fig 3G).

We next used chemical cross-linking coupled to mass spectrometry (XL-MS) to build a more comprehensive understanding of the association between Pch2 and ORC. XL-MS can provide information on inter- and intramolecular interactions that can yield useful insights into assembly principles of complex protein preparations. Using an experimental pipeline based on an MS-cleavable chemical cross-linker (DSBU, disuccinimidyl dibutyric urea, also known as BuUrBu) (Pan et al, 2018) (Fig 4A), we cross-linked purified Pch2 (His–MBP–Pch2) and ORC (His–Orc1-6) (Fig 4B) and after processing and MS analysis, identified cross-linked peptides (for cross-links, see Tables S1 and S2). (Note that DSBU is able to cross-link lysine, serine, and threonine residues.) We validated the quality of our XL-MS dataset by analyzing (intramolecular) cross-linked peptides within the MBP moiety present in our Pch2 preparation (see the Materials and Methods section for more detailed information). After applying a stringent cut off analysis by setting a false-discovery rate (FDR) of 2% (Pan et al, 2018), we obtained a total of 313 nonredundant cross-links out of a total of

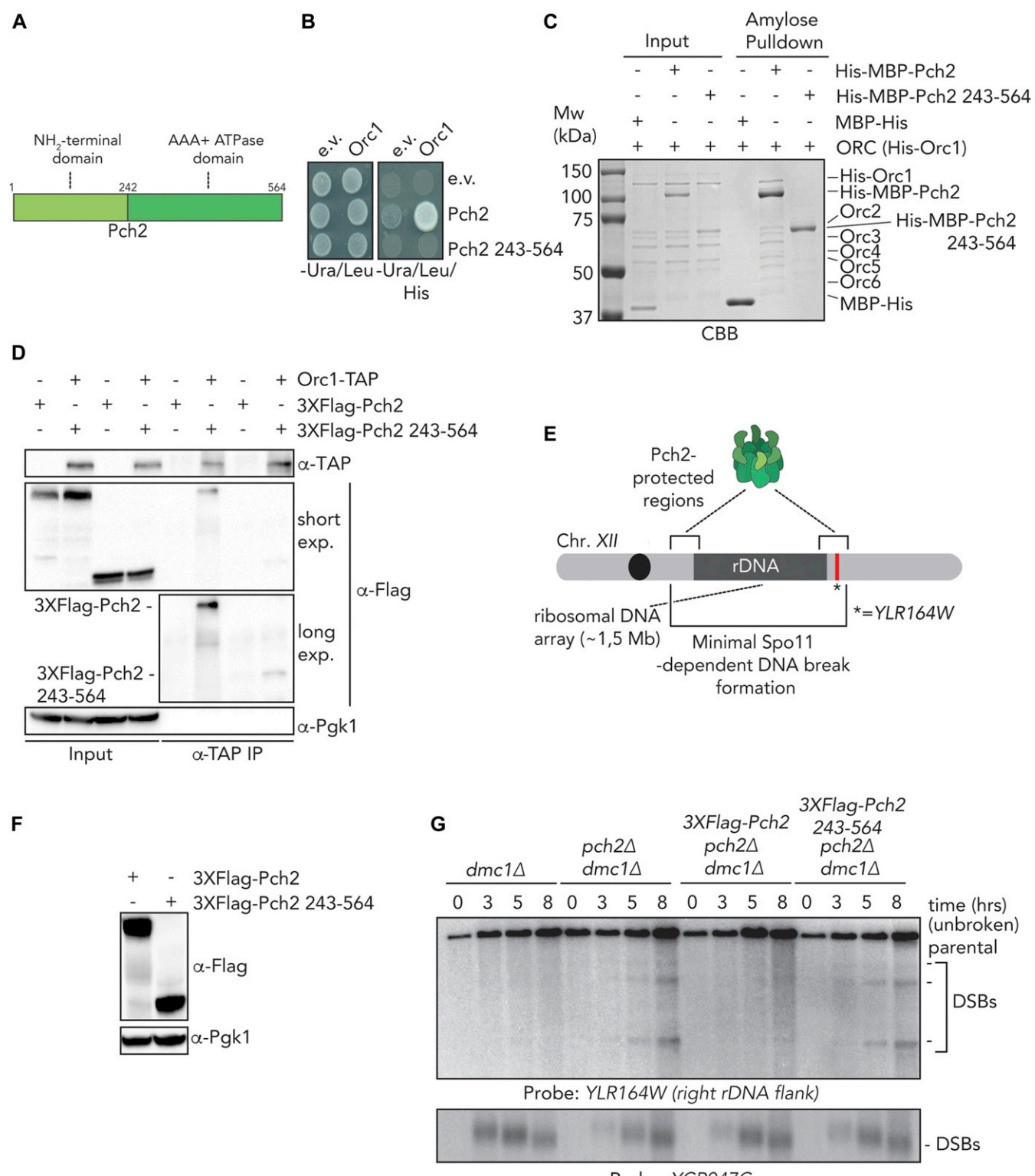

**Figure 3. The NH$_2$-terminal domain (NTD) of Pch2 is required for ORC–Pch2 formation.**
**(A)** Schematic of Pch2 domain organization. **(B)** Yeast two-hybrid analysis between Orc1 and Pch2 (full-length Pch2, and AAA+ ATPase domain [Pch2-243-564]).
**(C)** Amylose-based pulldown of His–ORC (His–Orc1-6) purified from insect cells, with His–MBP–Pch2 or His–MBP–Pch2-243-564. Coomassie Brilliant Blue (CBB) staining.
**(D)** Co-immunoprecipitation of 3xFlag-Pch2 and 3xFlag-Pch2 243-564 with Orc1–TAP (via α-TAP-IP) during the meiotic prophase (4 h into meiotic program). For α-Flag, short and long exposures are shown. α-Pgk1 is used as a loading control. **(E)** Schematic of the role of Pch2 in controlling Spo11-dependent DNA double-strand break (DSB) formation within the flanking regions of the budding yeast ribosomal DNA array located on chromosome *XII*. * indicates location of *YLR164W* locus, where DSB formation is

721 cross-linked peptides identified by MeroX (Fig 4C and Table S2). We used these nonredundant cross-links to generate cross-link network maps for the ORC–Pch2 assembly by using xVis (https://xvis.genzentrum.lmu.de) (Grimm et al, 2015). These 313 cross-links consist of 121 intermolecular cross-links (i.e., cross-links between peptides originating from two different proteins) and 192 intramolecular cross-links (i.e., cross-links between peptides originating from a single protein). We identified 96 Pch2–Pch2 cross-links (Fig 4C, red lines on Pch2 schematic, Fig S5 and Table S2). Importantly, because Pch2 forms a homo-hexamer, we cannot distinguish whether Pch2–Pch2–cross-linked peptides originate from intra- or intermolecular cross-linked peptides. We observed 77 cross-links between ORC subunits (i.e., inter-ORC cross-links) (Fig 4C, represented by blue lines. See also Fig S5A and B and Table S2). When comparing cross-link abundance between individual ORC subunits with a published cryo-EM structure of the ORC to model the position of each subunit (Fig 4F; based on structure protein data bank (PDB) 5v8f; [Yuan et al, 2017]), we noted that neighboring subunits often displayed the most abundant cross-links (e.g., Orc1/Orc2, Orc2/Orc3 and Orc3/Orc5; see Table S2). However, several observed cross-links span considerable distance when based on the ORC structure we used for analysis (PDB 5v8f; [Yuan et al, 2017]), arguing for significant levels of flexibility within our ORC preparation. Of note, our ORC is devoid of Cdc6, not associated with origin DNA, and also not bound to MCM–Cdt1, contrary to the reported structure (Yuan et al, 2017), which conceivably could affect complex topology. Furthermore, we cannot exclude that Pch2 leads to structural rearrangements of the ORC upon binding. A significant fraction of the cross-links (42 of 96; 44%) consisted of cross-links between peptides from the noncatalytic $NH_2$-terminal domain (NTD, amino acids 1–242) and the COOH-terminal AAA+ domain of Pch2 (amino acids 243–564) (Figs 4C and S5C). Because we cannot distinguish between inter- or intramolecular cross-links with respect to hexamer Pch2-derived peptides (see also above), these cross-linked peptides could be a reflection of (i) a close proximity between the NTD and AAA+ domain within a single Pch2 polypeptide or of (ii) an association between the NTD of one Pch2 monomer and the AAA+ domain of an adjacent (or potentially more distally localized, depending on domain flexibility) AAA+ module, from a distinct Pch2 monomer. With regard to these observations, we noted that in biochemical purifications, mutational disruption of the NTD of Pch2 influenced the apparent formation of stable/properly assembled Pch2 hexamers (Fig S4A), indeed hinting at a contribution of the NTD of Pch2 to the stable hexamerization of Pch2's AAA+ core. We also identified 21 inter-ORC–Pch2 cross-links (Figure 4C–F; black lines). Several observations are of note when considering these cross-links. First, we found cross-links that contain Pch2 peptides from both its enzymatic AAA+ core (12 out of 21; 57%) and its noncatalytic NTD (9 of 21; 43%) (see also Fig S5D and E). We interpret this to indicate that Pch2 makes extensive contacts with the ORC, whereby both its enzymatic core and its NTD come in close proximity to the

ORC. The observation that both Pch2's AAA+ core and NTD seem to be engaged in ORC binding is consistent with a similar scenario in which Pch2 binds to the ORC in an AAA+/client and/or adaptor-type engagement, as also indicated by our in vivo analysis (see above, Fig 1). The efficiency of such a binding mode is expected to rely on the hexameric state of Pch2. These results are in agreement with our in vivo observations using ATPase mutants (Fig 1). The largest fraction of the total Pch2–ORC cross-links is established between Pch2 and Orc1 (6 of 21; 29%). Orc1 is the largest subunit of the Orc1-6 complex, which might influence the distribution of the observed cross-links. However, the observed cross-links between Pch2 and Orc1 are all derived from a narrow region within Orc1 (spanning amino acids 551–615) located within the AAA+ ATPase core of Orc1. Thus, we interpret this enrichment for Orc1-derived cross-links to indicate that when associated with the ORC, Pch2 resides in close proximity to the Orc1 subunit. Combined with our in vivo observations, in which we consistently found Orc1 as the most robust factor associating with Pch2 (Fig 1), these findings suggest that Orc1 constitutes a major interaction hub for Pch2 within the ORC.

In addition to Orc1, Orc2 also forms a number of cross-links with Pch2 (Orc1/Orc2 together forms 10 of 21 cross-links; 48%). We noted that Orc1/Orc2 is neighboring the position that is occupied by Cdc6 when it is engaged with the ORC (Yuan et al, 2017). Cdc6 is not present in our preparations, leaving this space unoccupied. We thus speculate that Pch2 might use the vacated Cdc6-binding position to establish its interaction with the ORC. We attempted to map the 21 inter-ORC–Pch2 cross-links onto an ORC structure (PDB 5v8f [Yuan et al, 2017], Fig 4F; cross-linked residues are marked by a black dot). Because of the absence of regions of the ORC within the used structure, we were unable to map several of the ORC–Pch2 cross-links (i.e., cross-links with Orc2, Orc5, and Orc6; please refer to Fig 4D). Mapping of observed cross-links showed a distribution of residues across a large surface of the ORC, suggesting that Pch2 is in close proximity with a large region of the ORC when engaged. When we analyzed the position of these residues in a structure containing Cdc6, we noted that three cross-linked residues within Orc1 (K612, T614, and S615) were located in a position that is shielded by Cdc6 when bound to it, according to the ORC–Cdc6–Cdt1–MCM complex structure (PDB 5v8f [Yuan et al, 2017]). This finding reiterates the idea that Pch2 uses a binding mode that might involve binding interfaces within Orc1/ORC that is also involved in Cdc6 engagement during the G1 phase of the cell cycle. These data suggest that Pch2–ORC association can occur in the absence of Cdc6, and we performed in vivo several experiments to reinforce this premise. Pch2 expression is induced during the meiotic S-phase and G2/prophase, whereas Cdc6 availability is restricted to the G1 phase (Drury et al, 2000), also in the meiotic program (Phizicky et al, 2018). This mutually exclusive expression pattern already suggests that Pch2–ORC should not depend on Cdc6. We used a meiosis-specific null allele of *CDC6* (*cdc6-mn*) (Hochwagen et al, 2005) which interferes with premeiotic DNA replication (Fig S6A), to investigate if

interrogated. **(F)** Western blot analysis of meiotic time-course samples of yeast strains expressing wild-type 3xFlag-Pch2 and 3xFlag-Pch2 243-564 as used in (A). (*3xFlag-PCH2 pch2Δ dmc1Δ* and *3xFlag-pch2 243-564 pch2Δ dmc1Δ*). **(G)** Southern blot analysis of *YLR164W* locus (right ribosomal DNA flank; chromosome *XII*) and *YCR047C* locus (control DSB region; chromosome *III*), in *dmc1Δ*, *pch2Δ dmc1Δ*, *3xFlag-PCH2 pch2Δ dmc1Δ*, and *3xFlag-pch2 243-564 pch2Δ dmc1Δ* background. *dmc1Δ* is a DSB repair deficient mutant used to detect accumulation of meiotic DSBs.

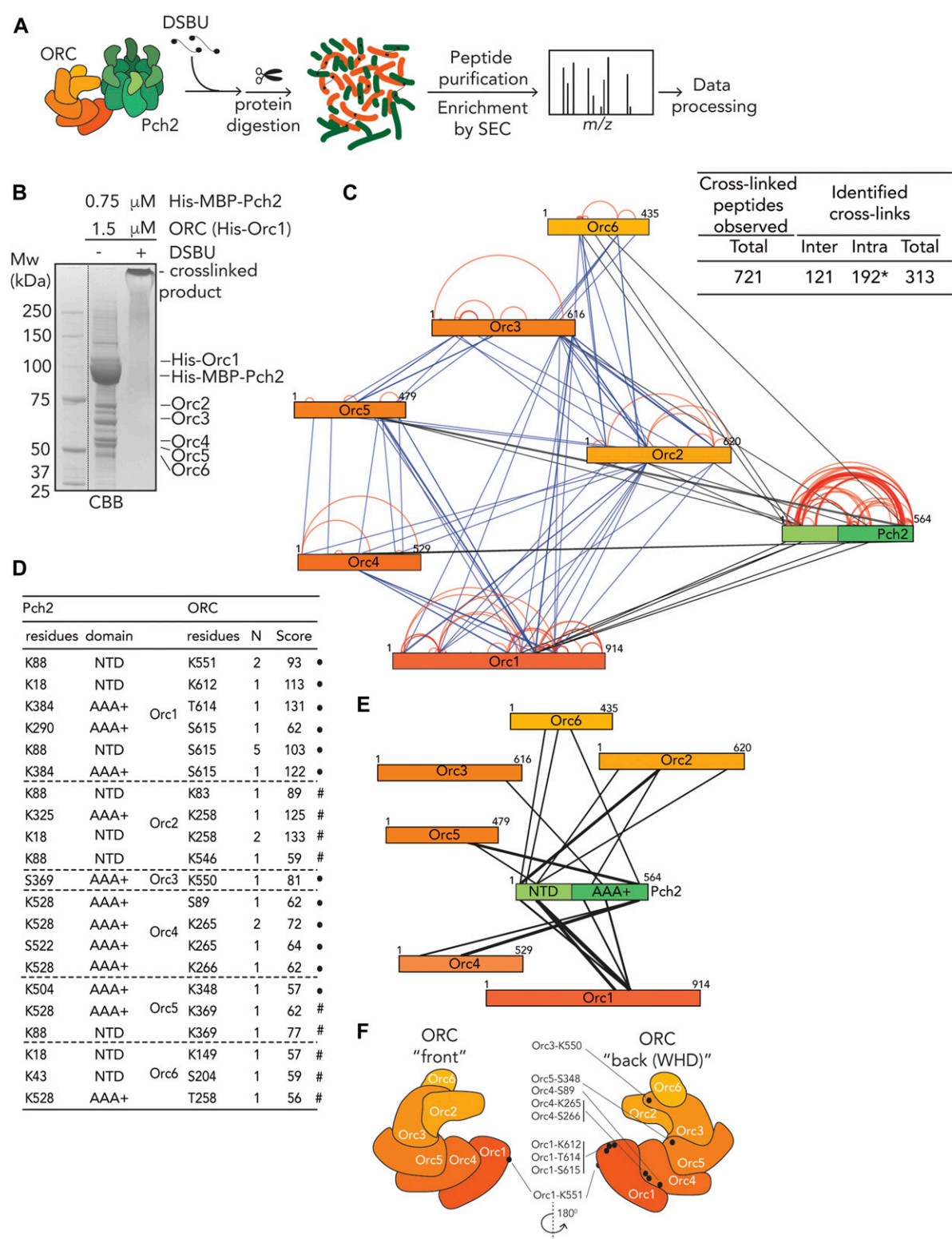

**Panel C table:**

| Cross-linked peptides observed | Identified cross-links | | |
|---|---|---|---|
| Total | Inter | Intra | Total |
| 721 | 121 | 192* | 313 |

**Panel D table:**

| Pch2 residues | Pch2 domain | ORC | ORC residues | N | Score | |
|---|---|---|---|---|---|---|
| K88 | NTD | Orc1 | K551 | 2 | 93 | • |
| K18 | NTD | | K612 | 1 | 113 | • |
| K384 | AAA+ | | T614 | 1 | 131 | • |
| K290 | AAA+ | | S615 | 1 | 62 | • |
| K88 | NTD | | S615 | 5 | 103 | • |
| K384 | AAA+ | | S615 | 1 | 122 | • |
| K88 | NTD | Orc2 | K83 | 1 | 89 | # |
| K325 | AAA+ | | K258 | 1 | 125 | # |
| K18 | NTD | | K258 | 2 | 133 | # |
| K88 | NTD | | K546 | 1 | 59 | # |
| S369 | AAA+ | Orc3 | K550 | 1 | 81 | • |
| K528 | AAA+ | Orc4 | S89 | 1 | 62 | • |
| K528 | AAA+ | | K265 | 2 | 72 | • |
| S522 | AAA+ | | K265 | 1 | 64 | • |
| K528 | AAA+ | | K266 | 1 | 62 | • |
| K504 | AAA+ | Orc5 | K348 | 1 | 57 | • |
| K528 | AAA+ | | K369 | 1 | 62 | # |
| K88 | NTD | | K369 | 1 | 77 | # |
| K18 | NTD | Orc6 | K149 | 1 | 57 | # |
| K43 | NTD | | S204 | 1 | 59 | # |
| K528 | AAA+ | | T258 | 1 | 56 | # |

**Figure 4. Cross-linking mass spectrometric analysis of origin recognition complex (ORC)–Pch2 complex assembly.**
**(A)** Schematic of DSBU-based cross-linking mass spectrometry (XL-MS) experimental pipeline. **(B)** Coomassie Brilliant Blue (CBB) staining of DSBU–cross-linked Pch2–ORC. **(C)** Right panel: Table indicating total cross-linked peptides and derived nonredundant (inter- and intramolecular) cross-links with a false discovery rate of 2%. * indicates that intramolecular cross-link peptides include 96 Pch2–Pch2 cross-links, which can be derived from inter- or intramolecular Pch2–Pch2 cross-links. Left panel: Schematic indicating all identified nonredundant cross-links. Blue: inter-ORC, red: intra-ORC and intra-Pch2, black: inter–ORC–Pch2. Network plots were generated using xVis. **(D)** Table showing inter–ORC–Pch2 cross-links. Indicated are residues in Pch2, and ORC subunits, domain of Pch2 involved (NTD: 1-242, AAA+: 243-564). N

depletion of Cdc6 influenced Pch2–ORC binding and function. (Note that in the *cdc6-mn* background, despite a failure to undergo bulk DNA replication [which indicates an efficient functional depletion], meiotic progression is unaffected and cells initiate DSB formation in a meiotic G2/prophase-like state [Hochwagen et al, 2005; Blitzblau et al, 2012].) As expected from our in vitro analysis, the interaction between Pch2 and Orc1 in the *cdc6-mn* background was similar to the binding observed in *CDC6* cells (Fig S6B), indicating that ORC–Pch2 assembly occurs under conditions of Cdc6 depletion. Similarly, Cdc6 depletion (via *cdc6-mn*) did not trigger a Pch2-like phenotype at rDNA borders, as judged by the analysis of meiotic DSB formation at the right rDNA flank (*YLR164W*) (Fig S6C). *pch2Δcdc6-mn* efficiently formed DSBs within the right rDNA flank (Fig S6C), demonstrating that bulk (Cdc6-dependent) DNA replication does not impact DSB formation in these regions, also in cells lacking Pch2.

Based on our earlier in vivo and in vitro results, we surmised that the NTD of Pch2 plays important roles in establishing the interaction between Pch2 and ORC. We aimed to further dissect this interaction. Based on Pch2 amino acid sequence conservation and secondary structure predictions (see Fig S7A for sequence alignments and placement of truncations), we performed Y2H analyses using a series of COOH-truncated fragments of Pch2. These analyses revealed that the NTD of Pch2 (consisting of amino acids 2–242) is sufficient to establish the interaction with Orc1 (Fig 5A). Further truncations of the NTD identified a minimal fragment of Pch2 (containing amino acids 2–144) sufficient for the interaction between Pch2 and Orc1 (Fig 5A, represented as a red dotted box). The behavior of this fragment is in agreement with our XL-MS analysis, which identified several cross-links between Pch2 and ORC subunits that consisted of Pch2 peptides that are located within this region of the NTD (K88, K18, K43; Fig 4D and E and Table S2). We attempted to express corresponding Pch2 NTD fragments in meiosis but often observed that these fragments were poorly expressed. This precluded us from performing in vivo interaction studies using these fragments. To further test a role for the NTD of Pch2 in mediating interaction with the ORC, we expressed recombinant NTD fragments. Similar to our in vivo observations, many recombinantly produced fragments were poorly expressed or aggregated under purifying conditions. We managed to express and purify the minimal NH$_2$-terminal fragment of Pch2 (His–MBP–Pch2-2-144) that was sufficient for Orc1 interaction in our Y2H analysis. SEC analysis suggested that this fragment exists as a monomer (expected size ~59 kD), which is in agreement with the crucial role AAA+ domains play in mediating hexamerization of AAA+ complexes (Fig 5B). This fragment was capable to interact with the ORC (Fig 5C and D). We noted however that this interaction is much weaker than the interaction with the full-length Pch2. This could indicate additional binding interfaces between Pch2 and ORC that lie outside of this domain (as suggested by the observation of additional

cross-links containing peptides from regions outside of the NTD of Pch2, and by the residual in vivo interaction we observed between Pch2–ΔNTD and Orc1; see above). Alternatively, this could indicate that hexamer formation of Pch2 (driven by AAA+/AAA+ interactions) is essential to increase the local effective concentration of the NTD. This would contribute to efficient binding between Pch2 and ORC. The latter interpretation is in agreement with our observation that the in vivo interaction between Pch2 and Orc1 is severely diminished in cells expressing a Pch2 Walker A domain mutant, which is expected to disrupt ATP binding and hexamerization (Herruzo et al, 2016) (Fig 1C). Altogether, these data strongly imply that the association of Pch2 with ORC constitutes a meiosis-specific assembly of two AAA+ protein complexes, which is dictated by unique interaction characteristics.

Canonical ORC function in yeast depends on ORC integrity and association with origins of replication (Foss et al, 1993; Loo et al, 1995; Fox et al, 1997; Suter et al, 2004; Shimada & Gasser, 2007). Our in vivo and in vitro analyses indicated that Pch2 interacts with the entire ORC. Pch2 is required to prevent rDNA-associated meiotic DSB formation (Vader et al, 2011). Inactivating Orc1 (via a temperature-sensitive allele of *ORC1*, *orc1-161*) triggers a similar rDNA-associated DSB formation as observed in cells lacking Pch2 (Vader et al, 2011). This rDNA-associated phenotype seen in the *orc1-161* background is exposed under conditions that are permissive for mitotic and meiotic DNA replication (i.e., they manifest at a permissive temperature of 23°C) (Vader et al, 2011), demonstrating a particular sensitivity of Pch2-associated phenotypes for Orc1 functionality. We sought to address whether a similar relationship existed between Pch2 and other ORC subunits. ORC subunits are essential for cell viability, and we used the "anchoraway" method (Haruki et al, 2008), which has been used to efficiently deplete chromosomal factors in budding yeast meiosis (Vincenten et al, 2015; Subramanian et al, 2016, 2019; Alfieri et al, 2018; Heldrich et al, 2020) to inactivate selected ORC subunits (Fig 6A). Mitotically proliferating cells that carry FRB-tagged versions of *ORC2* or *ORC5* (*orc2-FRB* and *orc5-FRB*) exhibited a strong growth defect when grown in the presence of rapamycin (Fig 6B), demonstrating efficient nuclear depletion. Because of technical reasons, we were unable to generate a functional *orc1–FRB* allele. To investigate the efficacy and timing of functional depletion, we used flow cytometry to query DNA replication in logarithmically growing cultures after treatment with rapamycin. In the *orc2–FRB* and *orc5–FRB* backgrounds, addition of rapamycin induced DNA replication to cease (as judged by an accumulation of *2C*-containing cells) within 180 min of treatment, with the first effects detectable after 90 min (Fig 6C). In addition, we confirmed nuclear depletion during meiosis via chromatin immunoprecipitation coupled to quantitative PCR (ChIP-qPCR) analysis on *orc2–FRB* cells (see below, Figs 6H and S9A and B). These experiments indicate a rapid

---

indicates how often cross-links were identified. MeroX score is indicated. ● indicates cross-linked ORC residues that are mapped into cartoon representation of the ORC structure in (F). # indicates cross-links that fall in regions of ORC subunits that are not present in the used ORC structure. **(E)** Schematic indicating identified nonredundant inter–Pch2–ORC cross-links. The line thickness corresponds to the number of cross-links, as shown in (D). **(F)** Cartoon depiction of ORC organization, based on structure PDB 5v8f; (Yuan et al, 2017). "Back" and "front" are relative to winged helix domain (WHD) orientation, as indicated. Black dots represent ORC cross-linked residues in our XL-MS analysis. Note that because of a lack of regions in the structure used to generate the ORC schematic representation, not all cross-links are represented (see also text).

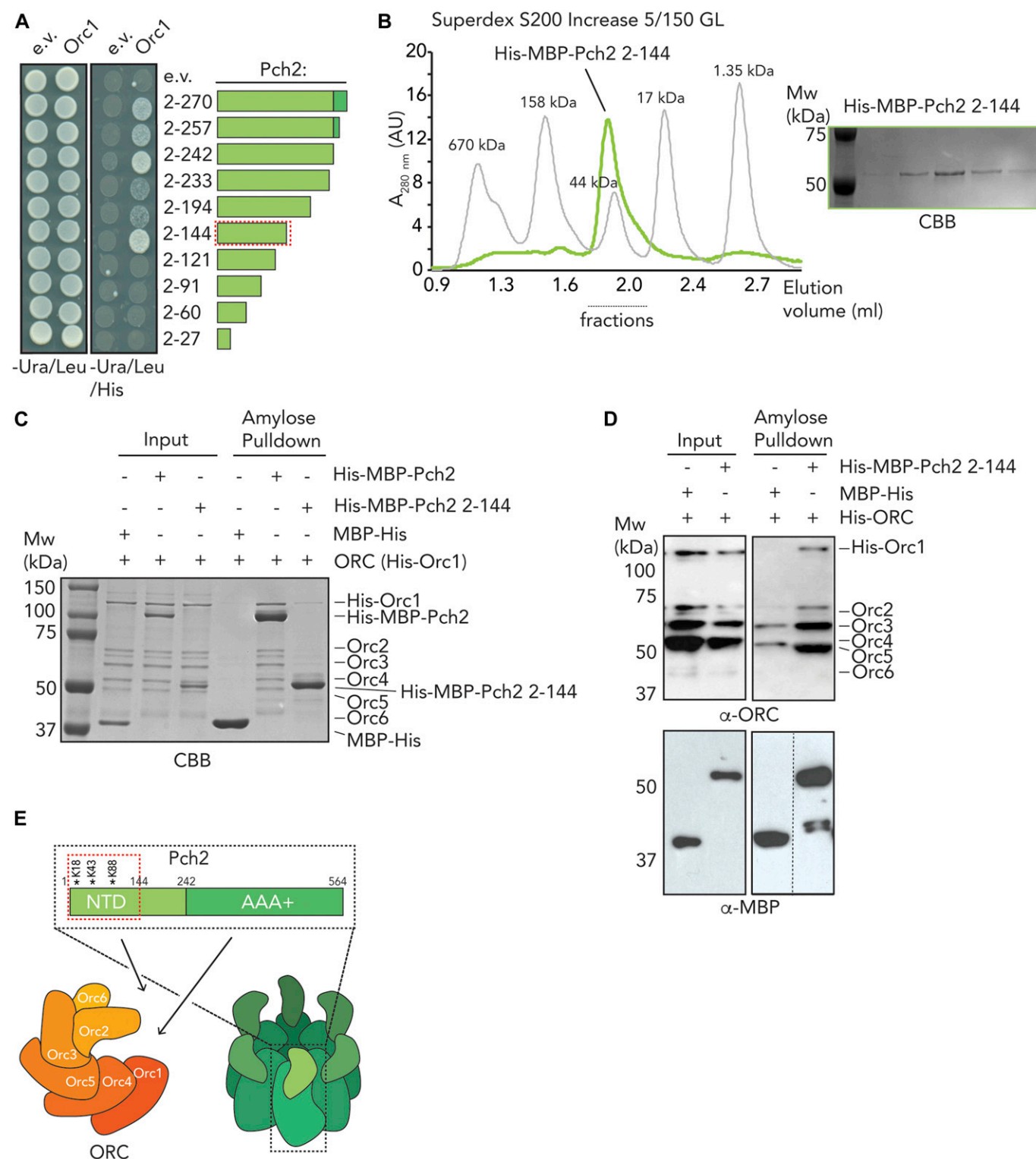

**Figure 5. Dissection of the role of the NTD of Pch2 in origin recognition complex (ORC) association.**
**(A)** Yeast two-hybrid analysis between Orc1 and NH$_2$-terminal fragments of Pch2 (2-270, 2-257, 2-242, 2-233, 2-194, 2-144, 2-121, 2-91, 2-60, 2-27). Red-dotted box indicates the minimal fragment of Pch2 that showed interaction with Orc1. **(B)** Size exclusion chromatography of His–MBP–Pch2-2-144 purified from insect cells; Coomassie Brilliant Blue (CBB) staining of the peak fractions (dotted line). AU, arbitrary units. **(C)** Amylose-based pulldown of the ORC (His–Orc1-6) purified from insect cells, with His–MBP–Pch2 or His–MBP–Pch2-2-144; Coomassie Brilliant Blue staining. **(D)** Amylose-based pulldown of the ORC (His–Orc1-6) purified from insect cells, with His–MBP–Pch2 or His–MBP–Pch2-2-144; Western blot analysis using α-MBP and α-ORC. **(E)** Schematic of interaction mode between the ORC and Pch2. Red-dotted box indicates the NH$_2$-terminal 2-144 region of Pch2's NTD. Cross-linked residues within the Pch2-2-144 region are indicated with *.

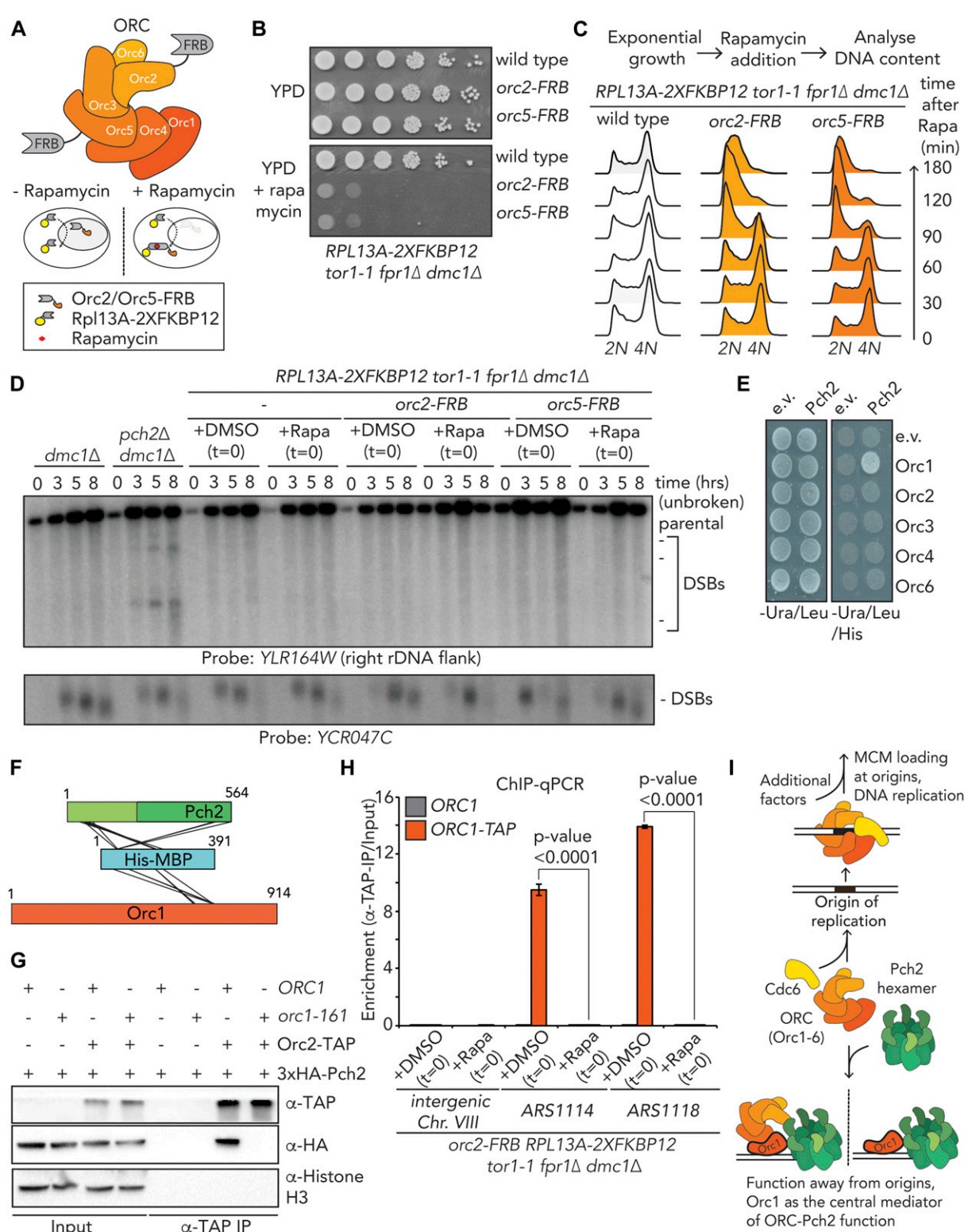

**Figure 6. Functional in vivo analysis of origin recognition complex (ORC)–Pch2.**

**(A)** Schematic of the ORC assembly and rapamycin-based anchor-away method. **(B)** 10-fold serial dilution spotting assay for anchor-away strains (untagged, *orc2-FRB* and *orc5-FRB*). Strains are grown on YP-dextrose (YPD) or YPD + rapamycin (1 μg/ml). **(C)** Flow cytometry analysis of efficiency of *orc2–FRB* and *orc5–FRB* nuclear depletion. Cells were treated as indicated, with rapamycin (1 μg/ml) at t = 0. **(D)** Southern blot analysis of *YLR164W* locus (right ribosomal DNA flank; chromosome *XII*) and *YCR047C* locus (control double-strand break [DSB] region; chromosome *III*). *dmc1Δ* is a DSB repair deficient mutant that is used to detect accumulation of meiotic DSBs. Rapamycin (1 μg/ml) or DMSO was added at indicated t = 0. Samples were taken at indicated time points after meiotic induction. **(E)** Yeast two-hybrid analysis between

functional depletion of Orc2 and Orc5. We used these *ORC* alleles to investigate rDNA-associated DSB formation (by probing meiotic DSB formation at the right rDNA flank; *YLR164W* [Vader et al, 2011]). Meiotic progression was normal under these conditions because meiotic DSB formation at a control locus (*YCR047C*; chromosome *III*) occurred normally (Fig 6D), and premeiotic DNA replication timing appeared unaffected under this treatment regimen. MCM association with origins of replication (the critical ORC-dependent step during DNA replication) occurs before induction into the meiotic program (and thus rapamycin exposure in our experimental setup) (Phizicky et al, 2018), and therefore, nuclear depletion of the ORC in this regimen is not expected to interfere with efficient premeiotic DNA replication. In contrast to what is observed in cells lacking Pch2 or in cells expressing *orc1-161* (Vader et al, 2011), rapamycin-induced depletion of Orc2 or Orc5 did not trigger an increase in rDNA-associated DSB formation (Fig 6D). Although we cannot exclude the possibility that Orc2/5 are incompletely depleted from the nucleus under the used conditions. The viability effects (Fig 6B), the timing of the observed effects on DNA replication of Orc2/5 depletion during vegetative growth (i.e., within 90–180 min; Fig 6C), and our ChIP-qPCR analysis (see below and Figs 6H and S9A and B) suggest that in our meiotic experiments (which include rapamycin treatments for 8 h), Orc2/5 are functionally depleted from the nucleus. Of note, experiments in which cells were exposed to longer periods of rapamycin treatment by adding the drug in premeiotic cultures (i.e., 3 h before initiation of meiotic cultures) also did not reveal an effect of Orc2/Orc5 depletion on rDNA-associated DNA break formation, despite the appearance of (mild) premeiotic DNA replication defects (unpublished observations, MA Villar-Fernández and G Vader). We conclude that under regimens that trigger clear replication defects in mitosis, we do not observe Pch2-like phenotypes in cells expressing depletion alleles of *ORC2* and *ORC5*. This is in striking contrast to phenotypes observed in cells expressing *orc1-161* under conditions (i.e., permissive temperature) where DNA replication is still supported in mitosis and meiosis, and Pch2-like phenotypes are exposed at the rDNA (Vader et al, 2011). These data suggest a specific reliance of Pch2 on Orc1 function that is not shared by Orc2 and Orc5, and together with several other observations (see below), point to Orc1 as a central functional interacting partner of Pch2. Orc1 was identified as an interactor of Pch2 via a Y2H screen (Vader et al, 2011), and when comparing Pch2 co-IP efficiencies of Orc1, Orc2, and Orc5, we consistently find the strongest interaction with Orc1 (Fig 1E). Our XL-MS analysis also hints at a crucial role for Orc1 as a Pch2 interactor (see above). We analyzed pair-wise interactions between individual ORC subunits (Orc1-4, and Orc6; Orc5 was not queried for technical reasons) and Pch2 using Y2H. We confirmed the interaction between Orc1 and Pch2, as reported earlier (Vader et al, 2011), but did not observe an interaction between Pch2 and Orc2-4 or Orc6 in this experimental

setup (Fig 6E). In addition, we analyzed our XL-MS dataset for intermolecular cross-links containing peptides from the MBP moiety (that is $NH_2$-terminally fused to Pch2 in His–MBP–Pch2). This revealed, in addition to 17 intermolecular cross-links between MBP and Pch2 (which are expected because these two polypeptides are covalently linked), 6 MBP–Orc1 intermolecular cross-links (Fig 6F and Table S2). In contrast, we observed no cross-links between MBP and other ORC subunits. Because efficient cross-linking depends on a proximity of ~12 Å between Cα's of cross-linked amino acids (Pan et al, 2018), these data argue that Orc1 is in close vicinity of MBP (and, by extension, Pch2). Together, these findings strengthen the idea that Orc1 plays a crucial role in mediating the ORC–Pch2 assembly. This predicts that the absence of Orc1 should lead to a decreased association between Pch2 and other ORC components. To test this premise, we probed the interaction between Pch2 and Orc2/Orc5 in the presence of a temperature-sensitive allele of *ORC1* (*orc1-161*). Indeed, Orc2 and Orc5 showed a decreased ability to immunoprecipitate Pch2 under such conditions, demonstrating that Orc1 is crucial in mediating the interaction between the ORC and Pch2 (Figs 6G and S8A). By contrast, the interaction between Orc1 and Pch2 was not affected by the nuclear depletion of Orc2 (in *orc2–FRB*–expressing cells) (Fig S8B).

The ORC function is closely linked to its association with origins of replication, and impairing ORC integrity (e.g., by removing one of its subunits) is expected to trigger its efficient removal from origins of replication. Because our data suggest that depletion of Orc2/Orc5 is not associated with Pch2-like phenotypes, one possibility is that efficient binding of Orc1/ORC to origins is not required for this function. We therefore tested the association of the ORC with origins of replication under the condition of nuclear depletion of Orc2. By performing chromatin immunoprecipitation coupled to quantitative PCR (ChIP-qPCR) of Orc1–TAP in cells expressing Orc2–FRB, we observed, upon the addition of rapamycin, a near-complete loss of Orc1 from selected euchromatic origins of replication (*ARS1114*, *ARS1118*, and *ARS1116*) in *orc2–FRB* cells (Figs 6H and S9A, see also Table S3). We also analyzed the effect of Orc2–FRB nuclear depletion on the association of Orc1 with the ARS localized at the rDNA (*ARS1216.5*). Although the loss of Orc1 from this rDNA origin was less dramatic than the effect that was seen at euchromatic origins, Orc1 binding at *ARS1216.5* was also significantly diminished upon rapamycin addition (Fig S9B). These results again underscore the nuclear depletion of Orc2 via the anchor-away method. In addition, because under identical treatment conditions, we do not observe rDNA-associated DSB formation (Fig 6D), these data argue that efficient binding of Orc1/ORC to origins is not strictly required for the function of Orc1/Pch2 in locally suppressing DSBs during the meiotic G2/prophase. In agreement with this, analysis of the chromosomal association of Pch2 (through immunofluorescence analysis of spread meiotic chromosomes) using identical depletion conditions showed that Pch2 localization was not significantly affected in

---

Pch2 and Orc1, Orc2, Orc3, Orc4, and Orc6. **(F)** Schematic indicating inter–MBP–Pch2 and inter–MBP–Orc1 nonredundant cross-links. **(G)** Co-immunoprecipitation assay of Pch2–E399Q with Orc2–TAP in *ORC1* or *orc1-161* backgrounds (via α-TAP-IP) during the meiotic prophase (4 h into meiotic program). Experiments were performed at 23°C. **(H)** TAP-based ChIP-qPCR in *ORC1* and *ORC1–TAP* expressing *orc2–FRB* anchor-away strains. Rapamycin (1 µg/ml) or DMSO was added at t = 0, and samples were taken at t = 4 h. Primers that amplify Intergenic *Chr. VIII* (control locus), *ARS1114*, and *ARS1118* were used. Experimental data are the average of three biological replicates. SEM is indicated. Significance was calculated using an unpaired *t* test, and *P*-values are indicated. **(I)** Model depicting the origin-independent function of Pch2–Orc1/ORC in local meiotic DSB control.

response to Orc2 or Orc5 depletion from the nucleus (nor specifically from the nucleolar/rDNA region) (Fig S9C–F).

Altogether, our data suggest that in vivo, Pch2 interacts with the entire ORC, with Orc1 being an important mediator of this interaction. Our findings suggest that Orc1 is a crucial functional mediator of Pch2's role at rDNA borders and argue that this function is executed away from origins of DNA replication and does not per se require Orc2/Orc5. We propose that during the meiotic G2/prophase, Orc1/ORC is repurposed and interacts with Pch2 in a biochemical and functional manner that is uniquely distinct from its well-documented roles at origins of replication (Fig 6I).

## Discussion

Pch2 is an important regulator of meiosis, and it uses its enzymatic activity to influence the chromosomal association of its clients. Up until now, HORMA domain–containing proteins were the only known direct interaction partners of Pch2/TRIP13 (Chen et al, 2014; Ye et al, 2015). Here, we demonstrate the direct association of Pch2 with a multi-protein complex essential for DNA replication (ORC). The canonical role for the ORC lies in forming the loading platform for the MCM replicative helicase (Bell & Kaguni, 2013; Bell & Labib, 2016), while it is also involved in gene silencing and sister chromatid cohesion (Foss et al, 1993; Loo et al, 1995; Fox et al, 1997; Suter et al, 2004; Shimada & Gasser, 2007). All these functions rely on association of the ORC with defined genomic regions, called origins of replication.

Our data reveal several biochemical characteristics about the ORC–Pch2 assembly and suggest that Pch2 uses an AAA+ /client and/or adaptor binding mode toward the ORC: (i) binding is increased in a mutant that stalls ATP hydrolysis (Fig 1B), (ii) hexamerization is required for efficient interaction (Fig 1C), and (iii) the nonenzymatic NTD of Pch2 plays a crucial role in mediating the interaction between Pch2 and ORC (Figs 3 and 5). In ORC–Cdc6, binding of a monomer of the Cdc6 AAA+ protein to the five other AAA+-like ORC proteins (Orc1-5) establishes the functional ring-shaped ORC hexamer (i.e., a Cdc6–Orc1-5 hexamer), which, in this composition, is proficient in loading the MCM AAA+ hexamer (Bell & Kaguni, 2013; Bell & Labib, 2016) (Fig 1D). An intriguing possibility regarding the association between Pch2 and ORC was that a monomer of Pch2 AAA+ protein could, in lieu of Cdc6, establish a complex with Orc1-5 (i.e., a 1:5 Pch2/Orc1-5 hexamer). However, we do not find evidence supporting such a binding mode. First, when we reconstituted the ORC–Pch2 complex, we observed that the pool of Pch2 that elutes at the expected size of a Pch2 hexamer interacts with the ORC (as judged by SEC analysis; Fig 2D). Second, the non-AAA+ domain of Pch2 (i.e., its NTD) provides a key contribution to the efficient binding of Pch2 to ORC (Figs 3–5). This would not be expected if a 1:5 Pch2/ORC (Orc1-5) would be established via binding principles that are similar to Cdc6–ORC, wherein AAA+/AAA+ interactions are the main driver of complex formation. Third, a Walker A domain Pch2 mutant that is expected to form a monomer (Herruzo et al, 2016) (Fig 1C) fails to interact with the ORC in vivo. Although we cannot formally exclude the existence of a 1:6 Pch2/ORC complex that binds to a hexamer of Pch2 (in a manner analogous to a 1:6 Cdc6–ORC

[Orc1-6]: hexameric MCM assembly), we interpret our experiments to indicate that the ORC (Orc1-6) is complexed with a hexamer of Pch2. In addition, we also show that Pch2–ORC assembly does not require Cdc6 (or any other accessory factors). During the meiotic program, expression of Pch2 is induced during the S-phase and peaks during the G2/prophase, when the ORC is not complexed with Cdc6 (Drury et al, 2000; Phizicky et al, 2018). In line with a temporal separation of Pch2- and Cdc6-bound ORC, we found evidence from in vitro reconstitution that Pch2 might (partially) engage the binding pocket that in ORC–Cdc6 is occupied by Cdc6.

Could the ORC be considered a client or an adaptor of Pch2? We believe that the most parsimonious explanation of our current analysis (combined with our earlier observations, which revealed that Orc1 is required for the nucleolar localization and function of Pch2 [Vader et al, 2011]), is that Orc1 plays an adaptor-like role, and as such acts as a recruiter of Pch2. The only known Pch2/TRIP13 adaptor is p31comet, a HORMA domain–containing factor that is required for the recognition of Mad2 during spindle assembly checkpoint signaling. p31comet is likely not conserved in budding yeast (Vleugel et al, 2012; van Hooff et al, 2017). We speculate that the ORC might have evolved adaptor functions for Pch2 in budding yeast meiosis.

However, in contrast to the role of the ORC as a chromosomal loader of MCM, our data imply that such a loading function for Pch2 is fundamentally different. Our ORC depletion experiments suggest that the loading role of Orc1 can be executed under conditions where ORC integrity is compromised (i.e., by nuclear depletion of Orc2 or Orc5), which is associated with a significant dissociation of Orc1 from origins of replication. This implies that the role of Pch2 (and by extension Orc1) should be executed away from origins of replication. In agreement with such a model, we recently reported genome-wide localization experiments of Pch2 during the meiotic G2/prophase, which revealed distinct binding patterns (which depend on Orc1) but no significant association of Pch2 with origins of replication (Cardoso da Silva et al, 2020).

How can we explain the role of Orc1 in directing Pch2 function, even under conditions where depletion of other ORC subunits compromises ORC integrity and origin association? A possibility is that in vivo, Orc1 could exist in two distinct pools: one where it is complexed with Orc2-6 (i.e., ORC) and the other where it exists as a monomer. Conceivably, Pch2 could interact with both pools. Such a model would be in agreement with our observation that in vivo, Orc1 exhibits a strong association with Pch2 (Fig 1E). If Orc1 is the factor that provides the needed functionality to Pch2 (whether complexed with ORC or not), inactivating other ORC components (such as Orc2/Orc5) would not per se trigger Pch2-like phenotypes. In such a scenario, the other non-Orc1 subunits of the ORC (Orc2-6) could be considered as piggybacking along with Orc1 and would not play any active role in Pch2 action. However, there may be subtle or additional roles for the non-Orc1 subunits for the ORC, which are not exposed in our current experimental approaches, for example, in aiding in binding affinity or activation of Pch2. In any case, our findings point to a noncanonical role for Orc1/ORC in mediating the function of Pch2 during the meiotic G2/prophase.

Based on these observations and following a model in which Orc1 is crucial to recruit Pch2 to the defined chromosomal regions, Orc1 should contain a chromosome-binding activity that is required for recruitment of Pch2. Indeed, Orc1 contains a nucleosome-binding

module (a bromo-adjacent homology [BAH] domain) (Callebaut et al, 1999; Yang & Xu, 2013), whose deletion mildly affects DNA replication (Müller, 2010). We have previously shown that this domain is required for the rDNA-associated role of Pch2 (Vader et al, 2011). Recent work has further demonstrated a role for the BAH domain of Orc1 in rDNA-associated protection against meiotic DSB activity (De Ioannes, 2019). Interestingly, our recent analysis of Pch2 chromosomal localization revealed that the BAH domain of Orc1 provides a crucial contribution to the proper recruitment of Pch2 to meiotic chromosomes (Cardoso da Silva et al, 2020). Taken together, these findings highlight the importance of this domain for meiosis-specific functions of Orc1/ORC, in connection to Pch2.

In conclusion, we have used a combination of in vivo and in vitro analyses to reveal the establishment of a meiosis-specific AAA+ assembly between ORC and Pch2. Our findings broaden the list of known ORC interactors by revealing a novel direct binding partner of the ORC (i.e., Pch2). Similarly, we also uncover a hitherto unknown direct association partner for Pch2 besides the already described HORMA domain–containing proteins. We propose that the function of ORC/Pch2 is executed away from origins of replication and strongly relies on Orc1 (Fig 6I). We suggest that Orc1/ORC is important in mediating the recruitment of Pch2 to chromosomes, where it can act on its client protein Hop1 to control DSB activity. In the future, it will be interesting to establish the dynamic interplay between Pch2, Hop1, and Orc1/ORC.

By establishing an in vitro reconstituted assembly of Pch2 and ORC combined with in vivo analysis, we have shed light on an interaction between Pch2 and an AAA+ adaptor-like protein complex. Our experiments reveal characteristics of this assembly and highlight certain plasticity in the ability of the ORC to interact with distinct AAA+ proteins. Understanding the biochemical, structural, and functional connections between these two ATPases in more detail will be an avenue for future research.

# Materials and Methods

### Yeast strains

All strains, except those used for yeast two-hybrid analysis and ORC purification, are of the SK1 background. See Supplemental Data 1 for a description of genotypes of strains used per experiment.

### Yeast two-hybrid analysis

Pch2 (full-length and different truncations/mutants) and Orc1-Orc6 were cloned in the pGBDU-C1 or pGAD-C1 vectors. The resulting bait and prey plasmids were transformed into a yeast two-hybrid reporter strain (yGV864). Yeast two-hybrid (Y2H) spot assay was performed by spotting 5 μl of cultures at an optical density at 600 nm (OD600) of 0.5 onto -Ura-Leu plates (control) and -Ura-Leu-His (selective plate) and grown for 2–4 d.

### Yeast viability assays

For spotting assays, anchor-away strains were grown on YP-glycerol plates overnight at 30°C, transferred to YP-dextrose (YPD) plates,

and further grown overnight at 30°C. Cells were then inoculated into 15 ml YPD culture and incubated overnight at 23°C and 180 rotations/minute (rpm) shaking. The following morning, cells were diluted to a final OD600 of 0.4 and grown for 4 h at 30°C and 180 rpm shaking. Then, 5 ml of cells were harvested at 90$g$ for 3 min, washed in 1 ml H$_2$O, and resuspended in 500 μl H$_2$O. And then, 10 μl of 10-fold serial dilutions were prepared and spotted on YPD plates with or without 1 μg/ml rapamycin. Growth at 30°C was monitored for the following 2–4 d.

### Meiotic induction

Cells were patched from glycerol stocks onto YP-glycerol plates and grown overnight. Patched cells were transferred to YPD and further grown overnight. Cells were cultured in liquid YPD at 23°C overnight and diluted at OD600 0.3 into presporulation media (BYTA; phthalate-buffered yeast extract, tryptone, and acetate). Cells were grown in BYTA for 16–18 h at 30°C, washed twice in water, and resuspended in sporulation media (0.3% KAc) at OD600 1.9 to induce meiosis. Sporulation cultures were grown at 30°C (except for experiments involving temperature-sensitive strains, where strains were grown at the permissive temperature [23°C]). For time courses in which the anchor-away system was used, rapamycin (1 μg/ml) or DMSO was added at t = 0 h (in sporulation media). Time courses were conducted, and samples for flow cytometry and Western or Southern blots were taken at different time points. For Western blot analysis, samples were taken after 0, 3, and 4 or 5 h, whereas for FACS and Southern blot analysis, samples were typically taken after 0, 3, 5, and 8 h.

### Flow cytometry

Flow cytometry was used to assess synchronous passage through the meiotic program (as judged by duplication of the genomic content) and was performed as described (Vader et al, 2011). For analysis of rapamycin-induced phenotype, mitotic cultures were grown to saturation and diluted to OD600 1.0, and rapamycin was added. Samples for flow cytometry were taken at the indicated time points. A total of 10,000 cells were counted for each sample.

### Western blot analysis

For Western blot analysis, protein lysates from yeast meiotic cultures were prepared using trichloroacetic acid (TCA) precipitation and run on 8% or 10% SDS gels, transferred for 90 min at 300 mA, and blotted with the selected primary/secondary antibodies, as described (Vader et al, 2011).

### Southern blot analysis

For Southern blot assay, DNA from meiotic samples was prepared as described (Vader et al, 2011). DNA was digested with *HindIII* (to detect DSBs at the control *YCR047C* hot spot) or *ApaLI* (to monitor DSBs in the region of interest: right rDNA flank; *YLR164W*), followed by gel electrophoresis, blotting of the membranes, and radioactive (32P) hybridization using probes specific for *YCR047C* (chromosome *III; 209,361–201,030*) or *YLR164W* (chromosome *XII; 493,432–493,932*)

(for detection of DSBs in the hot spot control region or rDNA, respectively) (Vader et al, 2011). DSB signals were monitored by exposing an X-ray film to the membranes and further developed using a Typhoon Trio scanner (GE Healthcare) after 1 wk of exposure.

### In vivo co-IP

For IP assays, 100 ml meiotic cultures at OD600 1.9 were grown, harvested after 4.5 h (G2/prophase; unless otherwise indicated), washed with cold $H_2O$, and snap frozen. Acid-washed glass beads were then added, together with 300 $\mu$l of ice-cold IP buffer (50 mM Tris–HCl pH 7.5, 150 mM NaCl, 1% Triton X-100, 1 mM EDTA pH 8.0, with protease inhibitors), and the cells broken with a Fastprep disruptor (FastPrep-24; MP Biomedicals) by two 45-s cycles on speed 6. The lysate was subsequently spun 3 min at 200$g$ and the supernatant transferred to a falcon tube. The lysate was next sonicated for 25 cycles (30 s on/30 s off), in high power range, using a Bioruptor (Bioruptor-Plus sonication device; Diagenode), and then spun down 20 min at 20,000$g$. Supernatant was transferred to a new microcentrifuge tube, and 50 $\mu$l of input was taken. For $\alpha$-Flag/HA/TAP-based IPs, 1 $\mu$l of antibody ($\alpha$-Flag-M2 antibody; Sigma-Aldrich/$\alpha$-HA; BioLegend/$\alpha$-TAP; Thermo Fisher Scientific) was added to the lysate and rotated for 3 h. After the incubation step, 30 $\mu$l of Dynabeads protein G (Invitrogen, Thermo Fisher Scientific) was added and rotated overnight at 4°C. For $\alpha$-Orc2-based IPs, lysate was precleared with 10 $\mu$l of Dynabeads protein G for 1 h at 4°C. Lysate was then incubated with 2 $\mu$l of $\alpha$-V5 (IgG isotype control; Invitrogen) or 11 $\mu$l of $\alpha$-Orc2 (Santa Cruz Biotechnology) for 3 h at 4°C, followed by 3-h incubation with 25 $\mu$l of Dynabeads protein G. Suspensions were washed four times with ~500 $\mu$l of ice-cold IP buffer. In the last wash, beads were transferred to a new microcentrifuge tube. Then, 55 $\mu$l of loading buffer was added, boiled at 95°C, and run in an SDS gel. The inputs followed a TCA precipitation step. Briefly, 10% TCA was added and incubated for 30 min on ice. Pellets were then washed with ice-cold acetone, spun, and dried on ice, and further resuspended in TCA resuspension buffer (50 mM Tris–HCl 7.5, 6 M urea). After incubating for 30 min on ice, pellets were dissolved by pipetting and vortexing. Finally, 10 $\mu$l of loading buffer was added, and samples were boiled at 95°C and run in an SDS gel together with the IP samples. Note that for the experiments shown in Fig 3D, 50 ml of sporulation culture, instead of 100 ml, was collected to perform the IP protocol. For Fig S1B and C, DNA was digested by addition of 20 U Benzonase (Sigma-Aldrich) to half of a 200 ml sample before cell lysis, and co-IPs were performed essentially as described. A fraction of the total lysate was used to isolate DNA using TRIzol (Thermo Fisher Scientific), according to manufacturer's protocol. DNA was analyzed using standard DNA PAGE.

### Expression and purification of recombinant proteins in insect cells

Full-length Pch2 and its truncated versions were purified from insect cells. Specifically, fragments containing the coding sequences of Pch2 or its truncations, derived from codon-imized cDNA, were subcloned into a pLIB–His–MBP vector (kind gift of Andrea Musacchio [Max Planck Institute of Molecular Physiology],

derived from pLIB [Weissmann et al, 2016]), and further integrated into EMBacY cells via Tn7 transposition. Positive clones were identified by blue/white screening and subsequently transfected into Sf9 cells to produce baculovirus (according to previously described methods) (Trowitzsch et al, 2010; Wilde et al, 2014). Baculovirus was amplified three times in Sf9 cells and used to infect Tnao38 cells for protein production. Tnao38 cells infected with the corresponding baculovirus (at a 1:10 dilution of virus to culture) were grown for 48 h, and pellets from 2 liter cultures were harvested. Cell pellets were resuspended in lysis buffer (50 mM HEPES pH 8.0, 300 mM NaCl, 5 mM imidazole, 5% glycerol, 5 mM $\beta$-mercaptoethanol, 1 mM $MgCl_2$, Benzonase, supplemented with SERVA protease inhibitor mix SERVA and cOmplete mini, and EDTA-free protease inhibitor cocktail tablets Sigma-Aldrich) and lysed by sonication (Branson Sonifier 450). Sonicated cells were cleared by centrifugation 1 h at 100,000$g$ (4°C) and the supernatant filtered. Clear lysate was immediately passed through a 5-ml TALON Superflow cartridge (Takara Bio). After extensive washing with buffer A (50 mM HEPES pH 8.0, 300 mM NaCl, 5 mM imidazole, 5% glycerol, 5 mM $\beta$-mercaptoethanol, and 1 mM $MgCl_2$) and wash buffer (50 mM HEPES pH 8.0, 1 M NaCl, 5 mM imidazole, 5% glycerol, 5 mM $\beta$-mercaptoethanol, and 1 mM $MgCl_2$), protein was eluted with a gradient between buffer A and buffer B (50 mM HEPES pH 8.0, 300 mM NaCl, 400 mM imidazole, 5% glycerol, 5 mM $\beta$-mercaptoethanol, and 1 mM $MgCl_2$). The presence of protein was monitored by UV280 nm. Those fractions containing the protein of interest were pooled and incubated 30 min at 4°C with pre-equilibrated amylose resin (New England BioLabs) and eluted with elution buffer (30 mM HEPES pH 8.0, 500 mM NaCl, 3% glycerol, 2 mM TCEP, 1 mM $MgCl_2$, and 20 mM maltose). The eluted protein was concentrated using an Amicon-Ultra-15 centrifugal filter (MWCO 30 kD) (Merck Millipore), spun down 15 min in a benchtop centrifuge (4°C), and subsequently purified by SEC, by loading onto a Superose 6 Increase 10/300 GL (GE Healthcare) previously equilibrated in gel filtration buffer (30 mM HEPES pH 8.0, 500 mM NaCl, 3% glycerol, 2 mM TCEP, and 1 mM $MgCl_2$). The peak fractions were analyzed by SDS–PAGE, and those fractions corresponding to the protein of interest were collected and concentrated using a 30K Amicon-Ultra-4 centrifugal filter (in the presence of protease inhibitors). The concentrated protein was snap frozen in liquid $N_2$ and stored at −80°C until further use. Note that for purification of His–MBP–Pch2-243-564, buffers were adjusted to pH 7.6 instead of pH 8.0.

The His-tagged ORC was purified from insect cells. The multiple subunits of the ORC were cloned using the biGBac method described in Weissmann et al (2016). Briefly, the coding sequences of the individual ORC subunits (Orc1, Orc2, Orc3, Orc4, Orc5, and Orc6) were cloned into pLIB vectors, with the particularity that Orc1 coding sequence was subcloned into a pLIB vector containing a 6xHis tag. pLIB vectors of His–Orc1, Orc2, and Orc3 were subsequently cloned into a pBIG1a vector, whereas the pLIB vectors of Orc4, Orc5, and Orc6 were assembled into a pBIG1b construct. pBIG1a and pBIG1b constructs were used to transform EMBacY cells by Tn7 transposition, and the positive clones were used to generate baculovirus by transfection to Sf9 cells. After 4-d amplification of the baculoviruses, the supernatant of both viruses containing His–Orc1-3 and Orc4–Orc6, respectively, were used for protein expression. A 3 liter culture of Tnao38 cells was coinfected with the

two baculoviruses and 48 h post-infection, cells were harvested by centrifugation, washed once with PBS, and snap frozen. Cell pellets were resuspended in lysis buffer (50 mM HEPES 7.5, 300 mM KCl, 1 mM MgCl$_2$, 10% glycerol, 5 mM $\beta$-mercaptoethanol, 5 mM imidazole, Benzonase, protease inhibitors SERVA protease inhibitor mix and cOmplete mini, and EDTA-free protease inhibitor cocktail) and lysed by sonication. Lysed cells were harvested by ultracentrifugation 1 h at 100,000$g$ (4°C), and the supernatant was filtered and precipitated with 20% (NH$_4$)$_2$SO$_4$ on ice for ~45 min and recentrifuged. Clear lysate was affinity-purified by incubating it with cOmplete His-tag purification resin (Roche) for 2 h (4°C). After extensive washing with a 5–15 mM imidazole gradient in buffer A (50 mM HEPES–KOH 7.5, 300 mM KCl, 1 mM MgCl$_2$, 10% glycerol, and 5 mM $\beta$-mercaptoethanol), protein was eluted with elution buffer (buffer A supplemented with 300 mM imidazole). The eluted protein complex was concentrated using a 30K Amicon-Ultra-15 centrifugal filter, spun down 15 min at 20,000$g$ in a benchtop centrifuge (4°C), and loaded onto a Superose 6 Increase 10/300 GL column (GE Healthcare), previously equilibrated in gel filtration buffer (30 mM HEPES pH 7.5, 300 mM KCl, 5% glycerol, 1 mM MgCl$_2$, and 2 mM TCEP). Fractions were analyzed by SDS–PAGE, and those fractions containing His–ORC were concentrated using a 30 kD MWCO concentrator and flash-frozen in liquid N$_2$.

### Expression and purification of ORC from budding yeast cells

The ORC was purified from budding yeast (yGV3358) essentially as described by the Diffley laboratory (Yeeles et al, 2015). The ORC that was directly concentrated on calmodulin beads was used for ORC-based pulldowns.

### Expression and purification of Hop1

Hop1 was purified from bacterial cells. Briefly, the coding sequence of Hop1 was subcloned into a pET28a vector for expression of recombinant NH$_2$-terminally polyhistidine-tagged Hop1 (6xHis–Hop1). For protein expression, BL21 RIPL cells were transformed with the resulting vector and further used to inoculate 11 L of LB media, supplemented with kanamycin and chloramphenicol. Cultures were grown at 37°C with vigorous shaking until OD600 ~ 0.6–0.8. Protein expression was induced by addition of 0.25 mM IPTG overnight at 18°C. Cells were harvested by centrifugation at 400$g$ for 15 min and the pellet washed with PBS and immediately snap frozen. For protein purification, cell pellets were resuspended in buffer A (50 mM HEPES, pH 7.5, 300 mM NaCl, 5 mM imidazole, 10% glycerol, 0.05% Tween-20, and 5 mM $\beta$-mercaptoethanol) supplemented with Benzonase and protease inhibitors (1 mM PMSF and SERVA protease inhibitor mix). Cells were lysed using a microfluidizer (Microfluidizer M-110S; Microfluidics Corporation), centrifuged at 100,000$g$, 4°C for 1 h, and the lysate filtered. The clear lysate was first passed through a 5-ml TALON column (GE Healthcare). After extensive washing, protein was eluted with an imidazole gradient between buffer A and buffer B (buffer A supplemented with 400 mM imidazole). Eluate was pooled, diluted 2:1 in buffer A without NaCl and imidazole, and subsequently loaded into a heparin column (HiTrap Heparin 16/10; GE Healthcare), previously equilibrated with buffer C (20 mM HEPES, pH 7.5, 150 mM NaCl, 5 mM MgCl$_2$, 10% glycerol, and 5 mM $\beta$-mercaptoethanol). Protein was further eluted in a gradient between buffers C and D

(buffer C with 1 M NaCl), and fractions pooled and concentrated using a 30K Amicon-Ultra-15 centrifugal filter. Concentrated protein was spun down 15 min in a benchtop centrifuge (4°C) and immediately loaded onto a HiLoad 16/600 Superdex 200 column (GE Healthcare), pre-equilibrated in gel filtration buffer consisting of 20 mM HEPES pH 7.5, 300 mM NaCl, 5 mM MgCl$_2$, 5% glycerol, and 2 mM $\beta$-mercaptoethanol. Fractions were analyzed by SDS–PAGE, and those fractions containing 6xHis–Hop1 were concentrated with an Amicon-Ultra-15 concentrator (MWCO 30 kD), snap frozen, and kept at –80°C until further use.

### In vitro pulldown assays

For pulldown between His–Hop1 and His–MBP–Pch2, 7.5 $\mu$l of amylose beads (New England BioLabs), preblocked with 5% BSA, were incubated with 6 $\mu$M His–MBP or 1 $\mu$M His–MBP–Pch2 (assuming a hexamer of Pch2) for 1 h on ice in a final volume of 30 $\mu$l pulldown buffer (50 mM Tris pH 7.5, 50 mM NaCl, 10 mM imidazole, 10 mM $\beta$-mercaptoethanol, 0.1% Tween-20, and 1 mM MgCl$_2$) supplemented with 200 $\mu$M ATP or 200 $\mu$M ATP$\gamma$S when specified. Beads were then washed once with 100 $\mu$l pulldown buffer, and 6 $\mu$M Hop1 was added. As input, 6% of the final volume was taken. This reaction was incubated for 90 min on ice and next washed once with wash buffer (50 mM Tris pH 7.5, 200 mM NaCl, 10 mM imidazole, 10 mM $\beta$-mercaptoethanol, 0.5% Triton X-100 and 1 mM MgCl$_2$). Then, 20 $\mu$l loading buffer was added, samples boiled at 95°C, and supernatant transferred to a new microcentrifuge tube. Samples were analyzed by SDS–PAGE and stained with Coomassie Brilliant Blue (CBB).

For pulldowns between His–MBP–Pch2 and His–ORC, 5 $\mu$l of 5% BSA preblocked amylose beads were incubated with 6 $\mu$M His–MBP or 1 $\mu$M His–MBP–Pch2 (assuming a hexamer of Pch2) for 1 h on ice in a 30 $\mu$l final volume of pulldown buffer (30 mM HEPES pH 7.5, 150 mM NaCl, 10 mM imidazole, 10 mM $\beta$-mercaptoethanol, 0.1% Tween-20, and 10 mM MgCl$_2$). The pulldown reactions were washed twice with 200 $\mu$l of pulldown buffer, and 1 $\mu$M ORC was added. As input, 10% of the final volume was taken. This reaction was incubated for 90 min on ice and washed twice with 200 $\mu$l wash buffer containing 30 mM HEPES pH 7.5, 200 mM NaCl, 10 mM imidazole, 10 mM $\beta$-mercaptoethanol, 10% Triton X-100, and 10 mM MgCl$_2$. Inputs were diluted with pulldown buffer up to 10 $\mu$l, and then loading buffer was added. For the pulldown reactions, 20 $\mu$l of loading buffer was added. Samples were boiled at 95°C, and supernatant from pulldown reactions transferred to a new microcentrifuge tube. Samples were analyzed by SDS–PAGE and stained with CBB. Alternatively, the input/pulldown samples were analyzed by Western blotting as follows: half of the input/pulldown reactions were run on SDS–PAGE gel, transferred at 300 mA for 90 min and probed overnight with $\alpha$-MBP (1:10,000; New England BioLabs) or $\alpha$-ORC (1:1,000, kind gift of Stephen Bell), and subsequently developed using the corresponding secondary antibody. For pulldown with dephosphorylated His–ORC, 3 $\mu$M of His–ORC was dephosphorylated by $\lambda$-phosphatase treatment at 4°C overnight (see below) in a total volume of 30 $\mu$l. Then, 10 $\mu$l was taken and incubated with either 1 $\mu$M of His–MBP–Pch2 or 6 $\mu$M of His–MBP in pulldown buffer, and pulldown was performed similarly as described above.

Pulldowns with Pch2 fragments (His–MBP–Pch2-2-144/His–MBP–Pch2-243-564) were performed similarly, except that 6 $\mu$M of His–MBP–Pch2-2-144 was used (due to formation of monomer instead of hexamer in this fragment. See the Results section for further details). Note that for pulldown with His–MBP–Pch2-2-144 and His–ORC analyzed by Western blot, we used a twofold excess of His–MBP–Pch2-2-144 fragment as compared with the pulldown analyzed by CBB. Western blotting was performed similarly as detailed above, probing with $\alpha$-MBP (New England BioLabs) or $\alpha$-ORC (kind gift of Stephen Bell, MIT).

For pulldowns with the ORC purified from budding yeast, the ORC bound to calmodulin beads was transferred into a 1.5-ml reaction tube (40 $\mu$l of the beads solution/condition) and washed with pulldown buffer: (30 mM HEPES pH 7.5, 200 mM NaCl, 10 mM $\beta$-mercaptoethanol, 0.1% Tryton, and 2 mM $CaCl_2$ [supplemented or not with 1 mM $MgCl_2$]) with or without nucleotides (ADP, AppNHP, and ATP$\gamma$S [50 $\mu$M] or ATP [50 $\mu$M]). His–MBP (1 $\mu$l of the MBP solution [300 $\mu$M stock]) or His–MBP–Pch2 (1 $\mu$M of the protein solution [40 $\mu$M, assuming a Pch2 hexamer]) was added for 1 h on ice. Then, 8 $\mu$l of the sample was taken as input. Inputs were diluted with pulldown buffer up to 20 $\mu$l, and then 5× SDS loading buffer was added. The pulldown reactions were washed twice with 200 $\mu$l of pulldown buffer, briefly spun down, and supernatant removed. And then, 20 $\mu$l of 2.5× SDS loading buffer was added. Samples were boiled at 95°C, and supernatant from pulldown reactions transferred to a new 0.5-ml microfuge tube. Samples were analyzed by SDS–PAGE and stained with CBB.

### Lambda phosphatase treatment of purified His–ORC

For dephosphorylation of His–ORC purified from insect cells, 2 $\mu$M of His–ORC were incubated with $\lambda$-phosphatase (1:5, $\lambda$-phosphatase:His–ORC) in a total volume of 30 $\mu$l in a buffer containing 30 mM HEPES pH 7.5, 300 mM KCl, 5% glycerol, and 2 mM TCEP. The dephosphorylation reaction was supplemented with 10 mM $MnCl_2$ and incubated for 1 h at room temperature or overnight at 4°C. Then, 5× SDS loading buffer was added, and samples were analyzed by SDS–PAGE followed by CBB staining.

### Analytical SEC

Analytical SEC was performed on a Superose 6 5/150 GL column (GE Healthcare) connected to an ÄKTAmicro FPLC system (GE Healthcare). Proteins (1 $\mu$M His–MBP–Pch2, 3 $\mu$M His–ORC) were mixed in a total volume of 50 $\mu$l, incubated 2 h on ice, and spun down for 15 min at 20,000$g$ in a benchtop centrifuge (4°C) before injection. All samples were eluted under isocratic conditions at 4°C in SEC buffer containing 30 mM HEPES pH 7.5, 150 mM NaCl, 3% glycerol, 1 mM $MgCl_2$, and 2 mM TCEP, at a flow rate of 0.1 ml/min. Fractions (100 $\mu$l) were collected, and 20 $\mu$l were analyzed by SDS–PAGE and CBB staining.

For SEC profiles represented in Figs 2A, 5B, S3A, and S4A, the purified proteins were run as described above. Briefly, purified His–MBP–Pch2 (2 $\mu$M), His–MBP–Pch2-2-144 (6 $\mu$M), or His–ORC (6 $\mu$M) or His–MBP–Pch2-243-564 (2 $\mu$M) were diluted in SEC buffer (30 mM HEPES pH 7.5, 150 mM NaCl, 3% glycerol, 2 mM TCEP, and 1 mM $MgCl_2$) up to a volume of 50 $\mu$l, spun down 15 min at 20,000$g$ (4°C),

and immediately loaded into a Superose 6 Increase 5/150 GL column (for His–MBP–Pch2 and His–ORC) or into a Superdex 200 5/150 GL (for His–MBP–Pch2-2-144).

### Cross-linking mass spectrometry

Cross-linking mass spectrometry (XL-MS) was performed as described (Pan et al, 2018). Briefly, 0.75 $\mu$M of His–MBP–Pch2 was mixed with 1.5 $\mu$M of His–ORC complex in 200 $\mu$l of buffer (30 mM HEPES pH 7.5, 150 mM NaCl, and 2 mM TCEP) and incubated at 4°C for 90 min. DSBU (disuccinimidyl dibutyric urea—also known as BuUrBu-, Alinda Chemical Limited) was added to a final concentration of 3 mM and incubated at 25°C for 1 h. The reaction was stopped by adding Tris–HCl pH 8.0 to a final concentration of 100 mM and incubated at 25°C for an additional 30 min. Then, 10 $\mu$l of protein sample was taken before and after adding the cross-linker for analysis by SDS–PAGE. SDS–PAGE gel was stained with CBB. Cross-linked protein complexes were precipitated by adding 4 volumes of cold acetone (−20°C overnight) and centrifuged 5 min at 20,000$g$, and the pellet was dried at room temperature. Protein pellets were denatured in denaturation reduction solution (8 M urea and 1 mM DTT) for 30 min at 25°C. Cysteine residues were alkylated by adding 5.5 mM chloroacetamide and incubating for 20 min at 25°C. ABC buffer (20 mM ammonium bicarbonate pH 8.0) was added to reduce the final concentration of urea to 4M. Sample was digested by Lys-C (2 $\mu$g) at 25°C for 3 h, followed by overnight Trypsin (1 $\mu$g) digestion in buffer containing 100 mM Tris–HCl pH 8.5, and 1 mM $CaCl_2$ at 25°C. The digestion was stopped by adding TFA to a final concentration of 0.2%.

Resulting peptides after digestion were run in three independent SEC runs on a Superdex Peptide 3.2/300 column (GE Healthcare) connected to an ÄKTAmicro FPLC system (GE Healthcare). SEC runs were performed at a flow rate of 0.1 ml/min in buffer containing 30% acetonitrile and 0.1% formic acid. And 100 $\mu$l fractions were collected, and the same fractions from the three SEC runs were pooled, dried, and submitted to liquid chromatography-mass spectrometry/mass spectrometry analysis.

The liquid chromatography-mass spectrometry/mass spectrometry analysis was performed as previously reported using an UltiMate 3000 RSLC nano system and a Q-Exactive Plus mass spectrometer (Thermo Fisher Scientific) (Pan et al, 2018). Peptides were dissolved in water containing 0.1% TFA and were separated on the UltiMate 3000 RSLC nano system (precolumn: C18, Acclaim PepMap, 300 $\mu$m × 5 mm, 5 $\mu$m, 100 Å, separation column: C18, Acclaim PepMap, 75 $\mu$m × 500 mm, 2 $\mu$m, 100 Å; Thermo Fisher Scientific). After loading the sample on the precolumn, a multistep gradient from 5–40% B (90 min), 40–60% B (5 min), and 60–95% B (5 min) was used with a flow rate of 300 nl/min; solvent A: water + 0.1% formic acid; solvent B: acetonitrile + 0.1% formic acid. Data were acquired using the Q-Exactive Plus mass spectrometer in data-dependent MS/MS mode. For full-scan MS, we used a mass range of m/z 300–1,800, resolution of R = 140,000 at m/z 200, one microscan using an automated gain control (AGC) target of 3 × $10^6$, and a maximum injection time of 50 ms. Then, we acquired up to 10 HCD MS/MS scans of the most intense at least doubly charged ions (resolution 17,500, AGC target 1 × $10^5$, injection time 100 ms, isolation window 4.0 m/z, normalized collision energy 25.0, intensity threshold 2 × $10^4$, dynamic exclusion 20.0 s). All spectra were recorded in profile mode.

Raw data from the Q-Exactive Plus mass spectrometer were converted to Mascot generic file format. Program MeroX (version 1.6.6.6) was used for cross-link identification (Gotze et al, 2015). Combined MS data in Mascot generic file format and the protein sequences in FASTA format were loaded on the program, and MS spectra matching cross-linked peptides were identified. In the settings of MeroX, the precursor precision and the fragment ion precision were changed to 10.0 and 20.0 ppm, respectively. RISE mode was used, and the maximum missing ions was set to 1. MeroX estimates the FDR by comparison of the distribution of the cross-link candidates found using provided protein sequences and the distribution of the candidates found from decoy search using shuffled sequences. A 2% FDR was used as the cutoff to exclude the candidates with lower MeroX scores. The results of cross-link data were exported in comma-separated value format. Cross-link network maps were generated using the xVis website (https://xvis.genzentrum.lmu.de) (Grimm et al, 2015). Validation of the datasets was performed by identifying 13 intra-MBP cross-links and using a published crystal structure of MBP (PDB 1FQB, [Duan et al, 2001]) to map Cα-Cα distances between identified cross-linked amino acids. The average Cα–Cα was 14.41 Å, which is in good agreement with the Cα–Cα distance (12 Å) which the cross-linked state of DSBU is able to facilitate (Table S2).

### Chromatin IP-qPCR

For ChIP experiments, 100 ml SPO cultures (OD600 of 1.9) were harvested 4 h after entering meiosis. Cultures were cross-linked with 1% methanol-free formaldehyde (FA) for 15 min at room temperature. Cross-linking was quenched with 125 mM glycine. After a wash with ice-cold TBS, cells were snap frozen and stored at −80°C. Cells were resuspended in 600 $\mu$l of TAP ChIP buffer (25 mM Tris–HCl pH 8.0, 150 mM NaCl, 01% NP-40, and 1 mM EDTA, pH 8.0) supplemented with EDTA-free protease inhibitors (Roche), 1 mM PMSF, 1× SERVA protease inhibitor mix (SERVA), and 1 mM sodium orthovanadate, and broken with glass beads using a bead beater (FastPrep-24; MP Biomedicals) (two times 60 s, speed 6, incubated on ice for 5 min in between runs). Chromatin was sheared by sonication using a Bioruptor UCD 200 (Diagenode) (settings: 25 cycles of 30 s on/off, high power at 4°C). Lysates were centrifuged at 16,000$g$ for 10 min at 4°C. Input samples were taken. Then, 550 $\mu$l of cell lysates was preincubated with 1 $\mu$g of anti-TAP (Thermo Fisher Scientific) for 3 h at 4°C before overnight incubation under rotation with magnetic Dynabeads protein-G (Invitrogen). Beads were washed four times with buffer containing detergent and another time with the same buffer without detergent. Reverse cross-linking, proteinase-K, and RNase-A treatments and final purifications and elution were performed with ChIP and input samples as previously described in Blitzblau and Hochwagen (2013). For the rDNA-specific ChIP experiment shown in Fig S9B, chromatin was extracted as above and IP was performed with the following modifications: cells were resuspended in FA buffer (50 mM HEPES–KOH, pH 7.5, 140 mM NaCl, 1 mM EDTA, 1% Triton X-100, 0.1% sodium deoxycholate, and protease and phosphatase inhibitors). Suspension was then incubated with 30 $\mu$l IgG Sepharose (GE Healthcare) for 3 h at 4°C. Beads were twice washed with FA buffer, twice with FA buffer containing 500 mM NaCl, and twice

with Tris/EDTA (TE) buffer. Reverse cross-linking, proteinase-K and RNase-A treatments, and final purifications and elution were performed as above. ChIP and input samples were quantified by qRT-PCR on a 7500 fast real-time PCR machine (Applied Biosystems). The experiment shown on Fig 6H was performed using the CFX-Connect real-time PCR detection system (Bio-Rad). The percentage of ChIP relative to input was calculated for the target loci as well as for the negative controls. The enrichment was calculated using the ΔCt method: $1/(2^{\wedge}[Ct - Ct_{control}])$.

Primers that were used amplify for qRT-PCR were as follows:

*Intergenic chromosome VIII-forwar*d: 5′-GCTGCATTTCCCACCACGTC-3′
*Intergenic chromosome VIII-reverse*: 5′-GCATTTAACACGGGCCACCA-3′
*PPR1*-forward: 5′-AGAACGTCATCTCCGGAATCT-3′
*PPR1*-reverse: 5′-TGGGCACGATGAGAGAAAGT-3′
*ARS1116*-forward: 5′-AAGCTTTTCATCCCAGCAGA-3′
*ARS1116*-reverse: 5′-TTTTTGTCGTTGTTCGATTCA-3′
*ARS1118-forward*: 5′-CCCTGATTATGGAGTGATTTTC-3′
*ARS1118-reverse*: 5′-GGACCGTCTGAAGAGGTGAA-3′
*ARS1114-forward*: 5′-TGAGCGTTTCCTTTTAGAT-3′
*ARS1114-reverse*: 5′-GCAATTGTTCCATTTTCTCC-3′
5S-forward: 5′-TGCGGCCATATCTACCAGAAA-3′
5S reverse: 5′-CACCTGAGTTTCGCGTATGG-3′
*ARS1216.5*: 5′-CACCACACTCCTACCAATAACGG-3′
*ARS1216.5*: 5′-AAAGGTGCGGAAATGGCTGA-3′

Primer efficiencies (calculated using standard procedures) were as follows: *Intergenic chromosome VIII = 1.889*, *PPR1* = 1.998, *ARS1116* = 1.991, *ARS1118* = 1.995, *ARS1114* = 1,943, *5S* = 2.008, *ARS1216.5* = 1.951.

### Chromosome spreads

Chromosome spreads and quantification of 3XHA–Pch2 in *orc2–FRB* or *orc5–FRB* backgrounds (nucleolus and non-nucleolar) were performed as described in Cardoso da Silva et al (2020). Chromosome synapsis was detected using an antibody against Gmc2 (kind gift of Amy MacQueen, Wesleyan University) (Voelkel-Meiman et al, 2019).

# Supplementary Information

# Acknowledgements

This work was financially supported by the European Research Council (Starting Grant URDNA, agreement nr. 638197, to G Vader), a CAPES-Humboldt fellowship from the Alexander von Humboldt Foundation (agreement nr. 99999.000021/2016-04, to R Cardoso da Silva) and the Max Planck Society. We thank Andrea Musacchio (Max Planck Institute of Molecular Physiology, Dortmund, Germany) for ongoing support and for sharing unpublished reagents. We thank Andreas Brockmeyer and Franziska Müller (Max Planck Institute of Molecular Physiology, Dortmund, Germany) for technical and computational assistance in preparation of the XL-MS dataset. We thank

Jolien van Hooff, Geert Kops (Hubrecht Institue, Utrecht, The Netherlands), and Berend Snel (Utrecht University, Utrecht, The Netherlands) for help with sequence alignments. We thank Stephen Bell (MIT, Cambridge, USA), Christoph Kurat, Allison McClure, John Diffley (Francis Crick Institute, UK), and Amy MacQueen (Wesleyan University, USA) for sharing reagents and protocols. We thank Andreas Hochwagen (New York University, New York, USA), Adèle Marston (Wellcome Centre for Cell Biology, Edinburgh, UK), Hannah Blitzblau (Novogy, Cambridge, USA), and Arnaud Rondelet (Max Planck Institute of Molecular Physiology, Dortmund, Germany) for comments on the manuscript. We thank members of the Vader and Bird groups (Max Planck Institute of Molecular Physiology, Dortmund, Germany) for helpful discussions and comments on the manuscript.

## Author Contributions

MA Villar-Fernández: conceptualization, formal analysis, investigation, and methodology.
R Cardoso da Silva: conceptualization, funding acquisition, investigation, and methodology.
M Firlej: investigation.
D Pan: formal analysis, investigation, and methodology.
E Weir: investigation and methodology.
A Sarembe: investigation and methodology.
VB Raina: investigation.
T Bange: formal analysis.
JR Weir: conceptualization, investigation, and methodology.
G Vader: conceptualization, formal analysis, supervision, funding acquisition, investigation, visualization, methodology, project administration, and writing—original draft, review, and editing.

## Conflict of Interest Statement

The authors declare that they have no conflict of interest.

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
