## [Reviewer comments · Life Science Alliance]

Life Science Alliance

Biochemical and functional characterization of a meiosis-specific Pch2/ORC AAA+ assembly

María Villar-Fernández, Richard Cardoso da Silva, Magdalena Firlej, Dongqing Pan, Elisabeth Weir, Annika Sarembe, Vivek Raina, Tanja Bange, John Weir, and Gerben Vader

DOI: <https://doi.org/10.26508/lsa.201900630>

Corresponding author(s): Gerben Vader, Max Planck Institute of Molecular Physiology

Review Timeline:

Submission Date:	2019-12-18
Editorial Decision:	2020-01-19
Revision Received:	2020-07-21
Editorial Decision:	2020-08-10
Revision Received:	2020-08-13
Accepted:	2020-08-13

Scientific Editor: Shachi Bhatt

Transaction Report:

January 19, 2020

Re: Life Science Alliance manuscript #LSA-2019-00630-T

Dr. Gerben Vader
Max Planck Institute of Molecular Physiology
Department of Mechanistic Cell Biology
Otto-Hahn-Strasse 11
Dortmund 44227
Germany

Dear Dr. Vader,

Thank you for submitting your manuscript entitled "Biochemical and functional characterization of a meiosis-specific Pch2/ORC AAA+ assembly" to Life Science Alliance. The manuscript was assessed by expert reviewers, whose comments are appended to this letter.

As you will see, the reviewers think that your work is largely robust and well done. However, they also raise some concerns that need to get addressed. We would thus like to invite you to submit a revised version of your manuscript, addressing the reviewers' concerns. Importantly, the concerns regarding the IPs performed and the request to include ATP in the assays need your detailed attention. The ORC-Pch2 interaction should thus get analyzed in the presence of Mg and ATP, ADP, ATPγS e.g. by pulldown. Further, ars1116 being a weak ORC binding site, other origins should get tested: eg., testing Orc1-Tap/Orc2-FRB and Orc1-FRB-TAP at other origins (e.g. ars1114/8) + localisation data (e.g. on meiotic spreads). Please also provide control data to exclude the possibility of a contamination in the purified ORC (reviewer point regarding Orc6).

Thank you for this interesting contribution to Life Science Alliance. We are looking forward to receiving your revised manuscript.

Sincerely,

B. MANUSCRIPT ORGANIZATION AND FORMATTING:

Reviewer #1 (Comments to the Authors (Required)):

The manuscript by Villar-Fenandez investigates whether a direct interaction between Pch2 and ORC functions during meiosis. Previous work, by Vader et al had shown that ORC and Pch2 function to stabilise the rDNA repeats during meiosis. Moreover, the past worked had shown that the BAH domain of Orc1 function in this process. Here, an in vivo TAP-IP approach has been used to determine the interaction of Orc1, Orc2 and Orc5 with HA tagged Pch2, which was supplemented with anti-Orc2 IP. The data showed a strong interaction between Orc1 and Pch2 and weaker interactions with Orc2 and Orc5. These interactions were greatly enhanced by a Pch2 Walker B ATPase mutant. In principle the interactions could have been mediated by DNA, as the IP was not treated with a nuclease. However, using purified proteins, the authors provided strong evidence that the ORC and Pch2 interact directly - using an IP approach, gel-filtration and cross-linking mass-spectrometry. Pch2 truncations were used to narrow down the interaction sites. The anchor-away approach was used to deplete Orc2 and Orc5 in vivo, which showed surprisingly no impact on rDNA repeat stability. Thus, the authors argue for an Orc1 independent function in meiosis.

In general, the manuscript is of high quality and covers an exciting topic, exploring an alternative function of ORC in meiosis. C-terminal tagging of Orc1, as done in Figure 1, can impact its function in helicase loading, so is not ideal. That said, it is possible that Pch2 contacts a different face of ORC than used for replication and Orc2 and Orc5 constructs and untagged ORC were used, so the evidence is very convincing. The purified Pch2 and ORC appear of good quality, although Orc6 appears to be the wrong size (Figure 2B and D). Alternatively, the band labelled Orc6 is a contamination, as the protein complex was not purified via the Mono Q column. Anyway, as Orc6 was detected by mass-spectrometry the construct is likely OK. I was surprised that in none of the experiments ATP was used. AAA+ ATPases are known to alter their conformation in the presence of ATP. Has this been tested at all? If not, it would be useful to discuss the matter. The crosslinking mass-spectrometry has shown a very strong N-term to C-term connection in Pch2. This indicates that the Pch2 organisational model presented here probably needs to be revised, as it is not fitting with the data. The N-C connection in Pch2 affects the interpretation of the Pch2 truncations, as truncations might disrupt the structure of the protein or of the hexamer. The authors performed gel-filtration on Pch2 (243-564) and reported that the protein adopts a more extended shape. I would like to see the data in a supplementary figure and wonder if the protein aggregated? Ideally, point mutants will be used to address the role of the Pch2 N-terminus in interaction with ORC. Apparently, these experiments have been performed, but the data are not shown. Is it possible to include any point mutant, as it would strengthen the understanding of the role of the Pch2 N-terminus in ORC binding? Alternatively, it would be best not to mention the point mutants at all. The minimal Pch2 fragment that interacts with Orc is Pch2 (2-144), although at reduced level. This makes sense, as probably multiple contacts are made. This is very convincing data.

Finally, the authors asked whether the entire ORC complex is important for protecting the rDNA borders from DSB during meiosis. Using the anchor away approach is in principle sensible, as it impairs DNA replication. I wonder, if the rDNA repeats, which have each an ORC binding site, are more resistant to ORC depletion - due to their high number of ORC binding sites. This could be addressed directly, by doing the positive control Orc1-FRB, which may produce different results than Orc1-161. If that is not possible the shortcoming should be at least discussed. In the abstract it is suggested that ORC function in meiosis independent of DNA binding? However, the anchor away approach removes ORC (including Orc1) via Orc2/Orc5 from the nucleus (Figure 6H). Either, there is a second pool of soluble Orc1, which acts independent of Orc2-6) or Orc1 would need to signal from the cytosol via Pch2 to rDNA repeats. This latter appears impossible. Alternatively, ORC or Orc1 binds to rDNA repeats in a more stable way than to ARS1116 (due to interaction with Pch2) or due to the Orc1-BAH domain - this possibility should be mentioned in the discussion.

Besides, the discussion could better connect previous results, which showed that the Orc1 BAH domain is essential in the process, to the current data. Also, the role of ATP-hydrolysis could be discussed. Pch2 ATP-hydrolysis appears to release ORC? Blocking Pch2 ATP hydrolysis via a Walker B mutant stabilises the ORC-Pch2 interaction, but still causes meiotic DSB at the rDNA repeat border. Thus, one could argue that not only complex assembly, but also release is probably important during the process?

Altogether, the study is performed to a very good standard. The results are very useful for researchers from many different fields - making this an attractive story. Minor comments below:

Introduction page 3 line 78. It would be useful to mention that the epigenetic marks are mostly used for human ORC, where sequence specificity is probably absent.

Results, page 17, line 508/509. The ORC structure used here (PDB 5v8f) is not a crystal structure, but a cryo-EM structure.

Reviewer #2 (Comments to the Authors (Required)):

The manuscript from Villar-Fernández et al. aims to characterize the nature of the Pch2/ORC assembly, two hexameric complexes predicated on AAA+ scaffolds. Previous work from the group has demonstrated that Orc1 is important for recruiting Pch2 to ribosomal DNA (rDNA) where Pch2 provides protection against Spo11-dependent double strand breaks and meiotic recombination. The present study implements in vitro and in vivo biochemical analysis to provide compelling evidence that Pch2 interacts directly with the ORC holocomplex (Orc1-6) and indicates that Orc1 may be the major interfacing subunit. Interestingly, the authors also find that ORC's role in Pch2-dependent rDNA protection does not rely on its ability to specify origins. This work constitutes a discrete step forward in our understanding of the Pch2/ORC assembly and prompts future investigations aimed at understanding how ORC directs the function of Pch2 at rDNA. There remain some issues with the manuscript that should be resolved prior to publication, including over-interpretation of some of the mass spec and in vivo pull-down assay data, and small technical issues.

Major points:

-Figure 4. A critical control for the mass spec study is to repeat the analysis with the ATP binding and hydrolysis defective mutants of PCH2 and see whether the cross-linking patterns stay the same or change as predicted from the interaction data in Figure 1. The resultant data may help resolve the extent to which interactions seen between Pch2 and subunits other than Orc1 are real.

-Throughout the work the in vivo pulldown assays are difficult to interpret, primarily owing to variable load levels and weak band intensity. The major issue arises from the authors' interpretation of Figure 1E, from which they conclude that Orc1 constitutes the major interaction hub for Pch2. While the results in Figure 1E clearly show that Orc1 IP results in the most significant enrichment of Pch2. This raises the question as to whether PCH2 is really interacting with the ORC hexamer. On the other hand, Supplemental Figure 1 shows the opposite: that IP of Orc2 and Orc5 results in higher levels of Pch2. Two possible ways to address this discrepancy include: 1) complete replicates of the in vivo pull-downs comparing Orc1/2/5, quantitate, and determine whether there exists a significant difference; or 2) use the pulldown assay only to make more general statements regarding the presence or absence of an ORC/Pch2 interaction.

-Pertaining to the above point, what accounts for the variability in the Histone H3 levels in Figure 1B? Is it correct to assume this is a load control? And what is the significance of the Pgk1 blot used throughout - another load control?

-Given the variability of the in vivo pulldown assays, the authors should provide further support for their conclusions with additional in vitro analysis of the Pch2/ORC complex. For example, the authors could express and purify the Pch2 E399Q and K320R mutant and assay the interaction with ORC in vitro to confirm their in vivo conclusions. The authors might also express and purify either Orc1 alone or Orc2-5 and complete in vitro pull down or analytical size exclusion experiments.

- There is no attempt to discern if ATP has an impact on the interactions observed by pull-down. Although the pull-downs do test ATP-binding mutants of Pch2, nucleotide should still be included as a comparison.

-Figure 5C: in the printed version, the ORC band intensity is too weak to evaluate whether there is, in fact, an interaction with Pch2-2-144. This assay should be repeated, and gel rerun with higher levels of ORC.

-Figure 4B: Why, if MBP-Pch2 (0.75 μ M) is at half the concentration of ORC (1.5 μ M), does the MBP-Pch2 band appear >10-fold more abundant than any Orc subunit band? Does this represent Pch2 hexamer concentration? And have the authors tried the crosslinking experiment with stoichiometric amounts of Pch2 hexamer:ORC hexamer, or with the complex formed by analytical sizing? If so, how do the results compare with the presented results?

- Figure 5A. The truncation points assigned to PCH2 seem rather arbitrary. Given that there is a structure of this protein and its N-terminal domain, where do the truncation points map? In particular, might residues 242 and 144 fall within linker regions, while residues 233 and 191 fall within a folded domain? This could explain why there is a peculiar loss of interaction, and then a recovery before it is lost again, as the protein is made progressively smaller.

- Figure 5C. Following from the prior point, it would seem important to repeat the pulldown essays with the slightly longer construct, e.g. the one ending at amino acid 242, since this may be better behaved?

-Figure 5C. Why does the fragment not pull down ORC1, yet pull down some of the partner proteins?

Minor points:

-Line74: the authors mistakenly state that "ORC is a universally conserved... complex". The ORC heterohexamer is, of course, unique to eukaryotes, and archaea often have only a reduced number of ORC homologs. There is no ORC homolog in bacteria.

figure 4D. Please indicate whether the interactions on ORC are to the AAA or WHD domains.

Figure 4E. Please indicate either by thickness or color whether each line refers to one, two, or five crosslinks as shown in panel 4D.

Figure 4F. The use of "front" or "back" is ambiguous. Please indicate whether you are referring to the face of ORC with the winged helix domains or not.

-PDB 5v8f is referenced as a crystal structure that, to the reviewer's understanding, is actually an electron microscopy structure

-For the Discussion: might the authors comment on the function of Pch2/HORMA domain-containing complexes and whether Pch2 interactions with HORMA domain proteins would be expected to be mutually exclusive with ORC interactions? This would actually be useful control study to carry out with the pulldown assays and reagents already in hand.

-The following grammatical issues were found:

Line58: missing "."

Line61: "of" should be moved within parentheses

Line74: "hetero-hexamer" should read "hetero-hexameric"

Line 93: references are incorrect. Please fix.

Lines 510 to 512: with the observed cross-links that don't match to PDB 5v8f match better to an alternate conformation of ORC such as PDB 4xgc or 5zr1?

Line541: "within the Orc1" should read "within Orc1"

Lines 552 to 554: please be more specific about which regions formed crosslinks but could not be mapped to the structure.

Line611: "interact" should read "interacts"

Line691: "function" should read "functional"

Reviewer #3 (Comments to the Authors (Required)):

In this manuscript, Ascensión Villar-Fernández and colleagues perform an extensive and rigorous characterization of PCH2's interaction with the ORC complex. Using a combination of in vivo and in vitro experiments, the authors map the interactions between PCH2 and the ORC complex to interrogate their roles in limiting double strand break formation at rDNA boundaries. They find that PCH2 interacts with all members of the ORC complex in vivo and in vitro. However, its interactions with Orc1 seem more robust than that with other Orc complex members. Further, removing *orc2* or *orc5* prevents Orc1 interaction with origins of replication without affecting double strand break formation near rDNA. This leads the authors to propose that Orc1 does not contribute to limiting double strand breaks near rDNA through its role at origins of replication and that Pch2's role in limiting double strand breaks acts only through Orc1 and not the other members of the Orc complex. A few additional experiments would provide further substantiation to their conclusions.

Major issues:

The last author previously showed that *orc1* temperature sensitive mutants at permissive temperature mislocalized Pch2. What is the localization of Pch2 on meiotic spreads in *orc2*-FRB and *orc5*-FRB? Further, what is the localization of ORC1-TAP in *orc2*-FRB and *orc5*-FRB? Is it present in the nuclei of meiotic cells, as predicted by their model? Does it colocalize with Pch2 on meiotic spreads?

Is there a reason the authors did not perform the anchors away experiment with Orc1 (Orc1-FRB)? Does it recapitulate the previously published *orc-1ts* experiment?

All of their in vitro interactions are between Pch2 hexamer and the entire Orc complex. In support of their model, do they detect an in vitro interaction between Orc1 and Pch2 hexamer, similar to the in vivo interaction they observe in *orc2*-FRB and *orc5*-FRB mutants? As an extension of this, is it

possible that the interactions mapped between Pch2 and other ORC complex members are the indirect result of Pch2's robust interaction with Orc1?

Minor

Please make clearer that the immunoprecipitation experiments in the first paragraph of the Results were performed in meiotic cells, and specifically at which point in meiosis. The materials and methods identify the 4.5 hour timepoint but for readers unfamiliar with yeast meiotic progress, this information will provide better context for these experiments. Referencing Supplementary Figure 1A here might also be useful.

Figure 1B: What is the role of anti-Pgk1 and anti-histone H3 westerns? To demonstrate equal loading? If so, not all input samples look the same.

Figure 1E: what are the additional bands that are present in the Orc1-TAP and Orc2-TAP anti-TAP IPs? They introduce confusion since they are running at molecular weights similar to other proteins tagged in the other samples (i.e. Orc2 and Orc5). Are they truncated products that are consistently seen? Indicating them on the figure and explaining them in the Figure Legend would be helpful.

Figure 3C: what fractions were used for the amylose pull down?

Can the authors discuss the inconsistency between yeast two hybrid and in vitro interactions in Figure 5?

Lines 670-672: "To test this premise, we probed the interaction between Pch2 and Orc2/Orc5 in the presence of a temperature sensitive allele of ORC1 (orc1-161)." For clarity, was this experiment performed at permissive or non-permissive temperature?

Point-to-point reply Villar-Fernandez *et al.*

Dear Editor,

We thank the reviewers for their constructive comments on our manuscript. In our revised manuscript, we have addressed concerns and questions, and outline detailed responses in our point-by-point reply below.

Comments from Editor:

As you will see, the reviewers think that your work is largely robust and well done. However, they also raise some concerns that need to get addressed. We would thus like to invite you to submit a revised version of your manuscript, addressing the reviewers' concerns. Importantly, the concerns regarding the IPs performed and the request to include ATP in the assays need your detailed attention. The ORC-Pch2 interaction should thus get analyzed in the presence of Mg and ATP, ADP, ATPγS e.g. by pulldown. Further, ars1116 being a weak ORC binding site, other origins should get tested: e.g., testing Orc1-Tap/Orc2-FRB and Orc1-FRB-TAP at other origins (e.g. ars1114/8) + localisation data (e.g. on meiotic spreads). Please also provide control data to exclude the possibility of a contamination in the purified ORC (reviewer point regarding Orc6).

To address the abovementioned concerns, the revised manuscript now includes:

- *In vitro* calmodulin pulldown experiment between CBP-ORC (purified from budding yeast, as described extensively by the Diffley lab, e.g. see (Yeeles et al., 2015)) and His-MBP-Pch2 in the presence/absence of MgCl₂, ADP, ATP and analogs (i.e. ATP/ATPγS /ADP/AppNHP) (Supplemental Figure 3H; see also Supplemental Figure 3F). Under these conditions, we do not detect noticeable differences in the binding between CBP-ORC and His-MBP-Pch2 in the presence of nucleotides, as compared with the pulldown without any nucleotides. We note that we have now also performed pulldown assays between His-MBP-Pch2 and the HORMA-domain protein Hop1 in the presence of ATP or ATPγS, which is a known substrate of Pch2 (Chen, et al., 2014) (Supplemental Figure 2B). Although (Chen, et al., 2014) showed that binding of Pch2 and Hop1 increases in the presence of ATPγS, we did not detect such a difference in our pulldown assays. We currently do not know why we do not detect differences in Pch2-ORC or Pch2-Hop1 under different ATP conditions, which is in contrast to what we have observed *in vivo* for Pch2-ORC (see Figure 1B and C) and what has been seen by others *in vitro* (Chen et al., 2014). One possible explanation, which we favour, is that the purified Pch2 used for the experiments in our manuscript (which is produced in insect cells) is lacking currently unknown post-translational modifications necessary for its ATP hydrolysis activity (of note, Chen *et al.*, purified GST-tagged Pch2 from yeast (Chen et al., 2014)), whereas we have purified His-MBP-Pch2 from insect cells). If the activity of His-MBP-Pch2 purified here is limited, the presence or absence of hydrolysable ATP is expected to have little effect on binding dynamics. We now mention this option for this difference between *in vivo* and *in vitro* observations regarding Pch2-ORC (and -Hop1) binding behaviour and its dependence on ATP hydrolysis in our manuscript. We note that the *in vitro* interaction assays between Pch2 and ORC in the presence of nucleotides included in the revised version of this manuscript, were performed with CBP-ORC purified from *S. cerevisiae* (instead of insect cells, as the His-ORC employed for the majority of pulldowns in this

- manuscript) due to a technical limitation in the expression/purification of His-ORC from insect cells. See also Supplementary Figure 3 for additional experiments using CBP-ORC and a comparison between ORC purified from insect cells (His-ORC) or budding yeast cells (CBP-ORC). We now also have mentioned the use of ORC from two different sources in our description of the data.
- ChIP-qPCR analysis in an *ORC1-TAP, orc2-FRB* strain at two additional euchromatic origins (*ARS1114* and *ARS1118*), as requested (Figure 6H). This analysis demonstrates that Orc1 binding to these ARSs is severely reduced upon the addition of rapamycin, demonstrating that the anchor-away technique used in this study is able to efficiently remove ORC from the origins of replication. We now also provide ChIP-qPCR data of the *ORC1-TAP, orc2-FRB* strain at the ARS within the rDNA (*ARS1216.5*) (Supplementary Figure 9B), which also shows a reduction of Orc1 association upon addition of rapamycin. We note that the depletion of Orc1 from the rDNA-origin is less stringent as compared to the loss of Orc1 at euchromatic origins. It is currently unclear whether this is a reflection of the repetitive nature of the rDNA.
 - Immunofluorescence analysis of 3HA-Pch2 in an *orc2-FRB/orc5-FRB* (Supplementary Figure 9C-E), which shows no significant difference in the Pch2 localization (either within the nucleolus, or on synapsing non-rDNA chromatin) upon rapamycin addition. Together with the performed ChIP analysis (see above), these data show that, under conditions where Orc1 is depleted from origin-associated sites upon Orc2/Orc5 nuclear depletion, Pch2 association with euchromatin or rDNA chromatin is not significantly impacted. Together with recent data from our lab (Cardoso da Silva et al., 2020), these data argue for a non-canonical role for Orc1 in mediating Pch2-chromosome association, in a manner that differs from the canonical function of Orc1 at origins of replication.
 - *In vitro* mass spectrometry data (Supplementary Figure 3C) showing that within the purified ORC (from insect cells) Orc6 is present, and that the purified ORC is indeed composed of all the ORC subunits (*i.e.* that it contains Orc1 through Orc6) (please refer to additional discussion in point 2 of Reviewer #1).

Reviewer's comments:

Reviewer #1 Comments to the Authors

1. The manuscript by Villar-Fenandez investigates whether a direct interaction between Pch2 and ORC functions during meiosis. Previous work, by Vader et al had shown that ORC and Pch2 function to stabilise the rDNA repeats during meiosis. Moreover, the past worked had shown that the BAH domain of Orc1 function in this process. Here, an in vivo TAP-IP approach has been used to determine the interaction of Orc1, Orc2 and Orc5 with HA tagged Pch2, which was supplemented with anti-Orc2 IP. The data showed a strong interaction between Orc1 and Pch2 and weaker interactions with Orc2 and Orc5. These interactions were greatly enhanced by a Pch2 Walker B ATPase mutant. In principle the interactions could have been mediated by DNA, as the IP was not treated with a nuclease.

We have now included a TAP immunoprecipitation (TAP-IP) assay in an *ORC1-TAP/3xHA-PCH2-E399Q* strain in both presence and absence of a nuclease (*i.e.* benzonase) to digest

DNA (Supplementary Figure 1B). This experiment shows that 3xHA-Pch2-E399Q and Orc1-TAP also interact when DNA is degraded, arguing against the possibility of this interaction being mediated by DNA. We provide evidence of the efficiency of the Benzonase treatment of the IP samples, as shown in Supplementary Figure 1C; *i.e.* upon Benzonase treatment, the nucleic acids in the samples are degraded and the nucleic acid smear disappears; compare lanes 1 and 2). We note that all TAP-IP experiments, as currently presented, include a rigorous sonication step to shear DNA (see also Material and Methods). In addition, in the majority of presented TAP-IP experiments, we include Histone H3 as a marker for chromatin, to ascertain a lack of chromatin contamination in our experiments.

2. However, using purified proteins, the authors provided strong evidence that the ORC and Pch2 interact directly - using an IP approach, gel-filtration and cross-linking mass-spectrometry. Pch2 truncations were used to narrow down the interaction sites. The anchor-away approach was used to deplete Orc2 and Orc5 in vivo, which showed surprisingly no impact on rDNA repeat stability. Thus, the authors argue for an Orc1 independent function in meiosis.

In general, the manuscript is of high quality and covers an exciting topic, exploring an alternative function of ORC in meiosis. C-terminal tagging of Orc1, as done in Figure 1, can impact its function in helicase loading, so is not ideal. That said, it is possible that Pch2 contacts a different face of ORC than used for replication and Orc2 and Orc5 constructs and untagged ORC were used, so the evidence is very convincing. The purified Pch2 and ORC appear of good quality, although Orc6 appears to be the wrong size (Figure 2B and D). Alternatively, the band labelled Orc6 is a contamination, as the protein complex was not purified via the Mono Q column. Anyway, as Orc6 was detected by mass-spectrometry the construct is likely OK.

We are confident that the ORC preparations we used contain all 6 subunits. Using SEC, we show that all ORC subunits co-migrate, suggesting stable complex assembly. We now provide the mass spectrometry data of ORC (purified from insect cells), which shows that all six ORC subunits can readily be detected in our purified complex (Supplementary Figure 3C). In addition, we have now also used purified (CBP-Orc1-tagged) ORC from *S. cerevisiae* (as described extensively by the Diffley lab, *e.g.* see (Yeeles et al., 2015)), and compared the molecular sizes of the subunits of ORC purified from insect cells and CBP-tagged ORC purified from *S. cerevisiae*, by performing SDS-PAGE followed by Coomassie Brilliant Blue (CBB) staining. As shown in Supplementary Figure 3D, the molecular sizes of ORC components purified from insect cells are comparable with those of the ORC subunits purified from budding yeast, indicating that the molecular size of the ORC components is correct. One observable difference between insect cell-purified ORC and budding yeast-purified ORC was the migration pattern of Orc6 (for example, see Supplementary Figure 3D): in the case of insect cell-purified ORC, Orc6 appeared to migrate as a double band, suggestive of phosphorylation that occurred during protein expression. Indeed, it is known that the Orc6 subunit of ORC can be phosphorylated (Nguyen et al., 2001, Weinreich et al., 2001). We find that this is also the case when ORC is purified from insect cells, by treating the purified protein with λ -phosphatase: under these conditions the double Orc6 band collapses into a single, faster migrating band, reminiscent of the Orc6 band that is seen in budding yeast purified ORC (Supplementary Figure 3E). We also include a binding assay comparing the ability of untreated insect cell-purified ORC (*i.e.* containing phosphorylated Orc6), and λ -phosphatase-treated ORC, demonstrating that phospho-status of ORC does not influence the binding of Pch2 with ORC. Finally, see also our response to general comments from the editor, where we elaborate further on the use of budding yeast-purified ORC to perform

additional binding assays with Pch2. Together these data show that Pch2 also interacts with budding yeast-purified ORC (Supplementary Figure 3).

3. I was surprised that in none of the experiments ATP was used. AAA+ ATPases are known to alter their conformation in the presence of ATP. Has this been tested at all? If not, it would be useful to discuss the matter.

These experiments have now been addressed (Supplementary Figure 2B and Supplementary Figure 3H). Please refer to the above reply to the general comments of the editor.

4. The crosslinking mass-spectrometry has shown a very strong N-term to C-term connection in Pch2. This indicates that the Pch2 organisational model presented here probably needs to be revised, as it is not fitting with the data. The N-C connection in Pch2 affects the interpretation of the Pch2 truncations, as truncations might disrupt the structure of the protein or of the hexamer. The authors performed gel-filtration on Pch2 (243-564) and reported that the protein adopts a more extended shape. I would like to see the data in a supplementary figure and wonder if the protein aggregated?

We agree with the reviewer that the extensive contacts between the NH₂-terminal and COOH-terminal domains of Pch2 might influence the interpretation of the data presented in our manuscript. Other AAA⁺ ATPases (such as the human Pch2 homolog, TRIP13; (Ye et al., 2017)) have been shown to establish contacts between the NH₂-terminal and C-terminal domains by similar cross-linking mass spectrometry (XL-MS) analysis as performed in our manuscript. Based on this, the current model is that the NH₂-terminal domain of TRIP13 is flexible and, upon interaction with partner proteins, changes the conformation or rotates in order to perform its biological function (Ye et al., 2017), similar to what has been observed for other AAA⁺ enzymes. It is quite conceivable that budding yeast Pch2 shares commonalities with TRIP13 regarding the ability of its NH₂-terminal domain to rotate and establish contacts with regions in the ATPase domain of the protein. As indicated in our manuscript, with the current XL-MS analysis provided here, it is not possible to distinguish whether the identified cross-links between the NH₂-terminal and C-terminal domains of Pch2 represent contacts between both domains of the same Pch2 monomer/hexamer or of two distinct Pch2 hexamers.

We have now included a SEC profile of purified His-MBP-Pch2-243-564 (Supplementary Figure 4A). When compared with the molecular weight standard, this purified protein elutes earlier than the full length His-MBP-Pch2 (see Figure 2A), suggestive of an extended shape of the purified ATPase domain. However, it is important to point out that this purified protein does not elute in the column void, as would be expected from a completely aggregated protein. Furthermore, when the purified protein is re-run on a SEC column, the elution pattern remains unchanged, again arguing against full unfolding and aggregation.

It is worth mentioning that point mutants in the N-terminal domain of Pch2 (His-MBP-Pch2-V5AD6A and His-MBP-Pch2-V9AR10A), also showed a similar SEC profile (not shown here). This suggests that indeed, the NH₂-terminal domain of Pch2 influences the shape/folding of this protein. We have decided to not include any data using the identified point mutants, see also point 5.

5. Ideally, point mutants will be used to address the role of the Pch2 N-terminus in interaction with ORC. Apparently, these experiments have been performed, but the data are not shown. Is it possible to include any point mutant, as it would strengthen the understanding of the role of

the Pch2 N-terminus in ORC binding? Alternatively, it would be best not to mention the point mutants at all.

We completely agree that such point mutants would be ideal. We have identified such point mutants. By performing Y2H assays with truncated mutants of Pch2 (based on sequence alignments and secondary structure prediction, see Supplementary Figure 7), we identified four residues in the most N-terminal region of Pch2 (V5, D6, V9, R10) that are highly conserved among *Saccharomyces*. Interaction between Orc1 and point mutants of such amino acids of Pch2 is disrupted by Y2H. In addition, these Pch2 point mutants are severely impaired in the ability to engage ORC *in vivo*, besides the similar expression levels of wild type Pch2 and Pch2 point mutants during meiosis. Southern blot assay shows that such Pch2 mutants behave as a *pch2Δ* or *pch2-ΔNTD* in terms of DSB formation at the budding yeast ribosomal DNA (rDNA) (presumably due to the lack of interaction of these mutants with Orc1/ORC). However, our preliminary *in vitro* experiments suggest that these mutations might cause a (partial) folding defect in Pch2. Because of this potential effect we cannot rule out that the observed *in vivo* phenotypes are an indirect cause of protein unfolding. We have therefore chosen to not include these *in vivo* experiments in our manuscript, and we have thus removed any mention of these mutants in our current manuscript.

6. The minimal Pch2 fragment that interacts with Orc is Pch2 (2-144), although at reduced level. This makes sense, as probably multiple contacts are made. This is very convincing data. Finally, the authors asked whether the entire ORC complex is important for protecting the rDNA borders from DSB during meiosis. Using the anchor away approach is in principle sensible, as it impairs DNA replication. I wonder, if the rDNA repeats, which have each an ORC binding site, are more resistant to ORC depletion - due to their high number of ORC binding sites. This could be addressed directly, by doing the positive control Orc1-FRB, which may produce different results than orc1-161. If that is not possible the shortcoming should be at least discussed.

The reviewer is correct to remark that, due to the more abundant binding sites of ORC within the rDNA (due to the presence of one ARS per rDNA repeat), the rDNA could be more resistant to ORC depletion via the anchor-away technique employed here. We are aware that the lack of effect in DSB formation at the rDNA in the *orc2-FRB* and *orc5-FRB* strains could be, to some extent, due to a partial removal of ORC upon rapamycin treatment. We agree with the reviewer that our manuscript would benefit from including Southern blot analysis with an *orc1-FRB* strain and comparing the phenotype (*i.e.* DSBs levels) with the *orc2-FRB*, *orc5-FRB*, and *orc1-161* alleles. However, our numerous attempts to generate an *orc1-FRB* allele failed, likely due to an incompatibility of *ORC1* functionality with FRB-based tagging. Having said that, we now have performed ChIP-qPCR analysis in the *orc2-FRB* strain at the rDNA (Supplementary Figure 9B). This analysis shows that the anchor-away technique is also able to displace Orc1 from rDNA-origins.

However, we also show that under the same conditions, Pch2 localization to the rDNA/nucleolus is not affected (Supplementary Figure 9C-E). See also above the reply to general comments from the editor. Furthermore, as suggested by the editor, we now provide ChIP-qPCR data at other ARSs (*ARS1118* and *ARS1119*) (Figure 6H). This data supports our observations that the anchor-away used in our study is able to remove ORC from selected ARSs (see also the reply to the comments of the editor). We however still cannot, as the reviewer rightly points out, fully exclude that incomplete depletion of ORC specifically from rDNA regions contributes to a lack of observed functional consequences. We mention this in fact in the manuscript. We do point out that, when using *orc1-161*, effects at the rDNA were

observed in meiosis already under conditions where premeiotic DNA replication is not affected (Vader et al., 2011). Taken together with these observations, our data using the *FRB*-tagged alleles suggest a specific function for Orc1/ORC at the rDNA that is not *per se* correlated to the level of total inactivation.

7. In the abstract it is suggested that ORC function in meiosis independent of DNA binding? However, the anchor away approach removes ORC (including Orc1) via Orc2/Orc5 from the nucleus (Figure 6H). Either, there is a second pool of soluble Orc1, which acts independent of Orc2-6) or Orc1 would need to signal from the cytosol via Pch2 to rDNA repeats. This latter appears impossible. Alternatively, ORC or Orc1 binds to rDNA repeats in a more stable way than to ARS1116 (due to interaction with Pch2) or due to the Orc1-BAH domain - this possibility should be mentioned in the discussion.

We believe that our data suggests that Pch2 recruitment to chromatin requires Orc1, but that this binding represents a non-canonical binding mode of Orc1/ORC to non-origin-associated regions. See also our recent report describing the binding patterns of Pch2 across (non-rDNA) chromatin (Cardoso da Silva et al., 2020). In that report, we indeed show that this association of Pch2 is helped by the BAH domain of Orc1. We now discuss these ideas and possibilities explicitly in the discussion. We also mention, in connection to Supplementary Figure 9B, that the depletion of Orc2 from the rDNA appears less penetrant as compared to the depletion we observed within euchromatic regions.

8. Besides, the discussion could better connect previous results, which showed that the Orc1 BAH domain is essential in the process, to the current data. Also, the role of ATP-hydrolysis could be discussed. Pch2 ATP-hydrolysis appears to release ORC? Blocking Pch2 ATP hydrolysis via a Walker B mutant stabilises the ORC-Pch2 interaction, but still causes meiotic DSB at the rDNA repeat border. Thus, one could argue that not only complex assembly, but also release is probably important during the process?

We now discuss the role of the BAH domain of Orc1 more explicitly, also referring to our recent study describing Pch2 chromatin association (Cardoso da Silva et al., 2020). We also more clearly discuss the role of the ATPase function of Pch2. We agree that our data argue that a dynamic binding (and release) of Pch2 to ORC is important for the function of Pch2 at the rDNA. The known role of Pch2 towards its HORMA domain-containing substrates (*e.g.* Hop1 in yeast) requires ATP hydrolysis (see (Chen et al., 2014, Ye et al., 2017)). The assembly of Hop1 onto meiotic chromosomes is required for efficient DSB formation, and Pch2 is known to remodel chromosomal Hop1, leading to its removal. As such, the most parsimonious explanation for the requirement of the ATP-activity of Pch2 to protect the rDNA is that this activity locally restricts to recruitment of Hop1.

Altogether, the study is performed to a very good standard. The results are very useful for researchers from many different fields - making this an attractive story. Minor comments below:

Introduction page 3 line 78. It would be useful to mention that the epigenetic marks are mostly used for human ORC, where sequence specificity is probably absent.

This has been now changed in the introduction (lines 74-80: “ORC binds to origins of replication, which in budding yeast are defined by a specific DNA sequence. Such sequence

specificity seems absent in *S. pombe* and metazoans, in which origins of replication are predominantly determined by chromatin structure, epigenetic marks and, specifically, the presence of a nucleosome-free region (Peng et al., 2015, Prioleau & MacAlpine, 2016).”)

Results, page 17, line 508/509. The ORC structure used here (PDB 5v8f) is not a crystal structure, but a cryo-EM structure.

This has been changed accordingly.

Reviewer #2 *Comments to the Authors:*

The manuscript from Villar-Fernández et al. aims to characterize the nature of the Pch2/ORC assembly, two hexameric complexes predicated on AAA+ scaffolds. Previous work from the group has demonstrated that Orc1 is important for recruiting Pch2 to ribosomal DNA (rDNA) where Pch2 provides protection against Spo11-dependent double strand breaks and meiotic recombination. The present study implements in vitro and in vivo biochemical analysis to provide compelling evidence that Pch2 interacts directly with the ORC holocomplex (Orc1-6) and indicates that Orc1 may be the major interfacing subunit. Interestingly, the authors also find that ORC's role in Pch2-dependent rDNA protection does not rely on its ability to specify origins. This work constitutes a discrete step forward in our understanding of the Pch2/ORC assembly and prompts future investigations aimed at understanding how ORC directs the function of Pch2 at rDNA. There remain some issues with the manuscript that should be resolved prior to publication, including over-interpretation of some of the mass spec and in vivo pull-down assay data, and small technical issues.

Major points:

1. Figure 4. A critical control for the mass spec study is to repeat the analysis with the ATP binding and hydrolysis defective mutants of PCH2 and see whether the cross-linking patterns stay the same or change as predicted from the interaction data in Figure 1. The resultant data may help resolve the extent to which interactions seen between Pch2 and subunits other than Orc1 are real.

We agree with the reviewer that it would in principle be interesting to perform the XL-MS analysis with the Pch2 ATP binding and hydrolysis defective mutants. We have now performed binding studies in the presence of ATP, ADP and non-hydrolysable forms of ATP (Supplementary Figure 3H), and also see general comments from the editor). Under these conditions, we do not detect observable differences in binding between Pch2 and ORC. Therefore, and in light of the significant technical and time investments such XL-MS experiments would require (also taking into account the current non-optimal work situation due to COVID-19), we have chosen to not perform these cross-linking experiments.

2. Throughout the work the in vivo pulldown assays are difficult to interpret, primarily owing to variable load levels and weak band intensity. The major issue arises from the authors' interpretation of Figure 1E, from which they conclude that Orc1 constitutes the major interaction hub for Pch2. While the results in Figure 1E clearly show that Orc1 IP results in the most significant enrichment of Pch2. This raises the question as to whether PCH2 is really interacting with the ORC hexamer. On the other hand, Supplemental Figure 1 shows

the opposite: that IP of Orc2 and Orc5 results in higher levels of Pch2. Two possible ways to address this discrepancy include: 1) complete replicates of the in vivo pull-downs comparing Orc1/2/5, quantitate, and determine whether there exists a significant difference; or 2) use the pulldown assay only to make more general statements regarding the presence or absence of an ORC/Pch2 interaction.

We assume the reviewer is referring to the co-immunoprecipitation (co-IP) assay presented in Figure 1B. We have now replaced this figure with a new co-IP assay, in which the loading levels of the controls used (Pgk1 and H3) are comparable among the samples (see also point 3 of Reviewer #2).

Although we agree with this reviewer that our co-IP experiment presented in Figure 1E does not *per se* prove that other subunits of ORC (besides Orc1), directly interact with Pch2, this experiment indicates that Pch2 is able to associate with Orc1 when it is complexed with other ORC subunits (Orc2 and Orc5) and presumably with the entire ORC during meiotic G2/prophase. This idea is reinforced with the co-IP assay presented in Figure 1F, which shows that Orc2 is able to immunoprecipitate 3xHA-Pch2-E399Q. Once more, this result does not prove a direct interaction between Pch2 and other ORC components besides Orc1, but provides evidence that Pch2 associates with Orc1 when it is complexed with other ORC subunits. Moreover, we have also probed the interaction between Pch2 and the entire ORC *in vitro* (Figure 2, and Supplementary Figure 3), which supports the idea that Pch2 is able to associate with the entire ORC. In addition, our XL-MS analysis also reveals contacts between Pch2 and all the ORC components (Figure 4 and Supplementary Figure 5D-E). Our hypothesis that Orc1 is the main interactor hub for Pch2 is substantiated by other pieces of evidence presented on this manuscript (see also Discussion). We have now specified this in the current version of the manuscript to avoid any possible confusion (lines 447-449: “We consistently observed a strong association between Pch2 and Orc1 relative to the other subunits tested in our comparative *in vivo* co-IP experiments (Figure 1E), which together with other data (XL-MS, co-IP assays, pulldowns and functional analysis; see below), suggests that Orc1 might be a central interactor of Pch2.”). Regarding the concern of this reviewer about Supplementary Figure 1, the experiment presented there is not a co-IP assay between Pch2 and the distinct ORC subunits, but a Western blot showing expression levels of these proteins at different times in meiosis.

3. Pertaining to the above point, what accounts for the variability in the Histone H3 levels in Figure 1B? Is it correct to assume this is a load control? And what is the significance of the Pgk1 blot used throughout - another load control?

Pgk1 and Histone H3 are used as loading controls. We have now included an improved experiment in Figure 1B, in which the loading controls of the inputs are comparable.

4. Given the variability of the in vivo pulldown assays, the authors should provide further support for their conclusions with additional in vitro analysis of the Pch2/ORC complex. For example, the authors could express and purify the Pch2 E399Q and K320R mutant and assay the interaction with ORC in vitro to confirm their in vivo conclusions. The authors might also express and purify either Orc1 alone or Orc2-5 and complete in vitro pull down or analytical size exclusion experiments.

We assume this reviewer is referring to the variability described in point 3, which we believe is caused by a misunderstanding; as stated above, Supplementary Figure 1A is not a co-IP, but a Western blot showing expression of either Orc1-TAP, Orc2-TAP or Orc5-TAP during meiotic progression. Moreover, the variability in the loading controls of Figure 1B has been solved by including an improved figure of the same experiment (see above). We have now performed binding studies between wild type Pch2 and ORC, in the presence of ATP, ADP and non-hydrolysable forms of ATP (Supplementary Figure 3H, and also see general comments from the editor). We agree with the reviewer that performing *in vitro* pull-down assays with Pch2-E399Q and Pch2-K320R would be interesting to pursue in the future. However, due to time restrictions, we have not been able to probe the interactions using such Pch2 mutants.

5. There is no attempt to discern if ATP has an impact on the interactions observed by pull-down. Although the pull-downs do test ATP-binding mutants of Pch2, nucleotide should still be included as a comparison.

We have now performed binding studies between wild type Pch2 and ORC, in the presence of ATP, ADP and non-hydrolysable forms of ATP (Supplementary Figure 3H, and also see general comments from the editor).

6. Figure 5C: in the printed version, the ORC band intensity is too weak to evaluate whether there is, in fact, an interaction with Pch2-2-144. This assay should be repeated, and gel rerun with higher levels of ORC.

We agree that the ORC intensity in the pulldown between ORC and Pch2-2-144 is weaker than that of ORC and full length Pch2. As indicated in the Result section (“This fragment was capable to interact with ORC (Figure 5C and D). We noted however that this interaction is much weaker as compared to the interaction with the full length Pch2. This could indicate additional binding interfaces between Pch2 and ORC that lie outside of this domain (as suggested by the observation of additional cross-links containing peptides from regions outside of the NTD of Pch2, and by the residual *in vivo* interaction we observed between Pch2- Δ NTD and Orc1; see above). Alternatively, this could indicate that hexamer formation of Pch2 (driven by AAA+ to AAA+ interactions) is essential to increase the local effective concentration of the NTD”) this result would be expected for such a truncated version of Pch2.

We have pursued to increase the intensity of the ORC bands seen by CBB staining, however, even after increasing the amounts of ORC used in the pulldown, His-MBP-Pch2-2-144 does not seem to be able to bind a higher amount of ORC that the already presented in Figure 5C. Since we are aware that the levels of ORC in such pulldown might be weak in the printed version, we performed a Western blot analysis of this pulldown assay (Figure 5D) that demonstrates that His-MBP-Pch2-2-144 is able to bind to ORC *in vitro*.

7. Figure 4B: Why, if MBP-Pch2 (0.75 μ M) is at half the concentration of ORC (1.5 μ M), does the MBP-Pch2 band appear >10-fold more abundant than any Orc subunit band? Does this represent Pch2 hexamer concentration? And have the authors tried the crosslinking experiment with stoichiometric amounts of Pch2 hexamer:ORC hexamer, or with the complex formed by analytical sizing? If so, how do the results compare with the presented results?

We believe there are several reasons why the Pch2 band appears so intense. First, the MBP-tag tends to be stained very efficiently by CBB. Second, the band of MBP-Pch2 is overlapping with His-Orc1. The reason that we add half amount of Pch2 hexamer (0.75 μ M) was that we were unsure whether Pch2 interacted with the ORC complex in a way of hexamer to hexamer or monomer to hexamer. So, we reduced the amount of Pch2. There might be some underestimation of the concentration of Pch2, which resulted in apparent larger amount judging by the SDS-PAGE. We agree with the reviewer that some cross-links might form due to the non-specific binding. However, we detected only a limited number of cross-links between Pch2 and ORC subunits on the limited surface areas of the ORC complex. We therefore believe it is unlikely that the excess of free Pch2 cross-linked to ORC non-specifically.

8. *Figure 5A. The truncation points assigned to PCH2 seem rather arbitrary. Given that there is a structure of this protein and its N-terminal domain, where do the truncation points map? In particular, might residues 242 and 144 fall within linker regions, while residues 233 and 191 fall within a folded domain? This could explain why there is a peculiar loss of interaction, and then a recovery before it is lost again, as the protein is made progressively smaller.*

As stated in our manuscript, the truncations of Pch2 were created based on protein sequence alignments among different organisms and secondary structure predictions. We have now included these alignments and secondary structure predictions in our manuscript (Supplementary Figure 7), and explicitly point to this figure for reference in the manuscript. All the truncated constructs of Pch2 mentioned by this reviewer (Pch2-2-242, Pch2-2-144, Pch2-2-233 and Pch2-2-191) showed an interaction with Orc1 via yeast two-hybrid. However, the difference in cell growth between some of these constructs should be interpreted with caution and we interpret the Y2H data only as an indication of interaction/no interaction, rather than in a quantitative manner (see also point 5 of Reviewer 3). For such analysis, it would be more accurate to perform quantification of either *in vivo* co-IP assays or *in vitro* pulldown experiments (see also point 9 of Reviewer 2) using these truncated versions of Pch2. Nonetheless, answering to this reviewer's concern, please refer to Supplementary Figure 7; none of the abovementioned residues (indicated with asterisks) fall within a folded protein region, at least as based on secondary structure predictions.

9. *Figure 5C. Following from the prior point, it would seem important to repeat the pulldown essays with the slightly longer construct, e.g. the one ending at amino acid 242, since this may be better behaved?*

We agree with this reviewer that it would be interesting to employ longer constructs of Pch2 in our pulldown assays. We have tried this, but observed that, *in vivo*, the NH₂-terminal domain of Pch2 (Pch2-2-242) was poorly expressed as compared to full length Pch2, which precluded us from expressing this truncated version of Pch2 *in vitro*. Our attempts to express and purify longer Pch2 constructs (in particular, His-MBP-Pch2-2-257 and His-MBP-Pch2-2-233) revealed that these truncated proteins form aggregates, as seen by size exclusion chromatography (SEC). This observation, together with the low amounts of "monomeric" protein obtained precluded us to perform interaction studies with longer Pch2 constructs.

10. *Figure 5C. Why does the fragment not pull down ORC1, yet pull down some of the partner proteins?*

As mentioned in Results, His-MBP-Pch2-2-144 is able to pulldown the Orc1 subunit (Figure 5C-D). We note that the molecular sizes of Orc1 and full length Pch2 (His-MBP-Pch2) are similar (see Figure 5C, lane 6), but it is clear that His-MBP-Pch2-2-144 is able to pulldown Orc1 (see Figure 5C, lane 7). This Pch2 fragment is not only able to pulldown the Orc1 subunit, but the entire ORC (see also Figure 5D, in which analysis of the pulldown via Western blot makes the binding more evident).

Minor points:

1. Line74: the authors mistakenly state that "ORC is a universally conserved... complex". The ORC heterohexamer is, of course, unique to eukaryotes, and archaea often have only a reduced number of ORC homologs. There is no ORC homolog in bacteria.

This has been changed accordingly in our manuscript.

2. Figure 4D. Please indicate whether the interactions on ORC are to the AAA or WHD domains.

Since we aim to focus our manuscript on Pch2, and the position of crosslinks/interactions within the Pch2 protein, we have chosen to not include additional information on the position of the crosslinked amino acids within the different ORC subunits. We note that the majority of the crosslinks identified do fall within the AAA+ core of the ORC subunits, especially in case of Orc1.

3. Figure 4E. Please indicate either by thickness or color whether each line refers to one, two, or five crosslinks as shown in panel 4D.

This has been done.

4. Figure 4F. The use of "front" or "back" is ambiguous. Please indicate whether you are referring to the face of ORC with the winged helix domains or not.

We have now indicated this in the Figure.

5. PDB 5v8f is referenced as a crystal structure that, to the reviewer's understanding, is actually an electron microscopy structure

This has been changed accordingly.

6. For the Discussion: might the authors comment on the function of Pch2/HORMA domain-containing complexes and whether Pch2 interactions with HORMA domain proteins would be expected to be mutually exclusive with ORC interactions? This would actually be useful control study to carry out with the pulldown assays and reagents already in hand.

Although we agree with the reviewer that it would be interesting to investigate whether the interaction of Pch2 and the meiotic HORMA domain-containing protein Hop1 is mutually exclusive with the interaction of Pch2 with ORC, we consider that such experiments are

beyond the scope of our manuscript, due to the current limited time available. However, such interaction assays would be interesting for future work.

-The following grammatical issues were found:

1. *Line58: missing "."*

This has been changed accordingly.

2. *Line61: "of" should be moved within parentheses*

This has been changed accordingly.

3. *Line74: "hetero-hexamer" should read "hetero-hexameric"*

This has been changed accordingly.

4. *Line 93: references are incorrect. Please fix.*

This has been corrected.

5. *Lines 510 to 512: with the observed cross-links that don't match to PDB 5v8f match better to an alternate conformation of ORC such as PDB 4xgc or 5zr1?*

PDB 4xgc is a structure of *Drosophila melanogaster* ORC, and 5zr1 is a structure of ORC bound to origin DNA. We consider that for XL-MS mapping between Pch2 and ORC, the PDB 5v8f structure is the most appropriate model to use.

6. *Line541: "within the Orc1" should read "within Orc1"*

This has been changed accordingly.

7. *Lines 552 to 554: please be more specific about which regions formed crosslinks but could not be mapped to the structure.*

We now have included an explicit statement regarding this in the legend of Figure 4.

8. *Line611: "interact" should read "interacts"*

This has been changed accordingly.

9. *Line691: "function" should read "functional"*

This has been changed accordingly.

Reviewer #3 Comments to the Authors:

In this manuscript, Ascensión Villar-Fernández and colleagues perform an extensive and rigorous characterization of PCH2's interaction with the ORC complex. Using a combination of in vivo and in vitro experiments, the authors map the interactions between PCH2 and the

ORC complex to interrogate their roles in limiting double strand break formation at rDNA boundaries. They find that PCH2 interacts with all members of the ORC complex in vivo and in vitro. However, its interactions with Orc1 seem more robust than that with other Orc complex members. Further, removing orc2 or orc5 prevents Orc1 interaction with origins of replication without affecting double strand break formation near rDNA. This leads the authors to propose that Orc1 does not contribute to limiting double strand breaks near rDNA through its role at origins of replication and that Pch2's role in limiting double strand breaks acts only through Orc1 and not the other members of the Orc complex. A few additional experiments would provide further substantiation to their conclusions.

Major issues:

1. The last author previously showed that orc1 temperature sensitive mutants at permissive temperature mislocalized Pch2. What is the localization of Pch2 on meiotic spreads in orc2-FRB and orc5-FRB? Further, what is the localization of ORC1-TAP in orc2-FRB and orc5-FRB? Is it present in the nuclei of meiotic cells, as predicted by their model? Does it colocalize with Pch2 on meiotic spreads?

We have now included microscopy data showing the localization of Pch2 in either *orc2-FRB* or *orc5-FRB* strains (in presence of DMSO (control) or the drug rapamycin) (Supplementary Figure 9C-E). We find that the localization of Pch2 on meiotic spreads is unaltered (whether it is within the nucleolus or with euchromatin) in such strains in the presence of rapamycin (*i.e.* nuclear depletion of either Orc2 or Orc5 does not lead to Pch2 loss). See also the reply to de editor's comments. We have also attempted immunofluorescence assays to visualize the localization of Orc1 in an *Orc1-TAP*, *orc2-FRB* strain. However, we encountered technical issues with the antibodies tested to detect Orc1 that prevent us from performing these experiments.

2. Is there a reason the authors did not perform the anchors away experiment with Orc1 (Orc1-FRB)? Does it recapitulate the previously published orc-1ts experiment?

See point 6 of Reviewer #1.

3. All of their in vitro interactions are between Pch2 hexamer and the entire Orc complex. In support of their model, do they detect an in vitro interaction between Orc1 and Pch2 hexamer, similar to the in vivo interaction they observe in orc2-FRB and orc5-FRB mutants? As an extension of this, is it possible that the interactions mapped between Pch2 and other ORC complex members are the indirect result of Pch2's robust interaction with Orc1?

We agree with the notion that Orc1 is likely a central mediator of the interaction between Pch2 and ORC, and it would indeed be interesting to test the binding of Pch2 with Orc1 *in vitro*. We have attempted to express and purify Orc1 in isolation in insect cells, but these efforts have not been successful, potentially due to protein unfolding of Orc1. As such, we have not been able to investigate the interaction between Pch2 and Orc1 *in vitro*. We note that to our knowledge, budding yeast Orc1 has not been successfully purified independently of other ORC subunits by other researchers.

Minor issues:

1. Please make clearer that the immunoprecipitation experiments in the first paragraph of the

Results were performed in meiotic cells, and specifically at which point in meiosis. The materials and methods identify the 4.5 hour timepoint but for readers unfamiliar with yeast meiotic progress, this information will provide better context for these experiments. Referencing Supplementary Figure 1A here might also be useful.

This has been changed accordingly in Materials and Methods (line 178, “harvested after 4.5 hours (meiotic G2/prophase; unless otherwise indicated”). We have also included the reference to Supplementary Figure 1 here.

2. Figure 1B: What is the role of anti-Pgk1 and anti-histone H3 westerns? To demonstrate equal loading? If so, not all input samples look the same.

Pgk1 and Histone H3 are indeed used as loading controls, and we have now included an improved experiment of Figure 1B (See also point 3 of Reviewer #2).

3. Figure 1E: what are the additional bands that are present in the Orc1-TAP and Orc2-TAP anti-TAP IPs? They introduce confusion since they are running at molecular weights similar to other proteins tagged in the other samples (i.e. Orc2 and Orc5). Are they truncated products that are consistently seen? Indicating them on the figure and explaining them in the Figure Legend would be helpful.

We thank the reviewer for pointing this out and apologize for this confusion. The bands to which the reviewer refers likely represent degradation products of either Orc1 or Orc2. This has now been properly indicated in Figure 1E, as well as in Figure 1E legends (*i.e.* degradation products are denoted with asterisks).

4. Figure 3C: what fractions were used for the amylose pull down?

As indicated in Materials and Methods, the inputs represent 10% of the total pulldown volume (30 μ L). The total volume of both inputs and pulldown reactions were loaded in an SDS-PAGE and stained by CBB.

5. Can the authors discuss the inconsistency between yeast two hybrid and in vitro interactions in Figure 5?

We assume that the reviewer is referring to the lesser extent of interaction between ORC and Pch2-2-144 observed via pulldown (as compared with full length Pch2, Figure 5C), whereas via Y2H this fragment of Pch2 seems to interact with Orc1 to a good extent (Figure 5A). As mentioned in the Results, the fragment Pch2-2-144 constitutes the minimal fragment able to sustain interaction with Orc1/ORC and the lesser extent of interaction between this fragment and ORC (via pulldown assay, Figure 5C) is probably due to the presence of additional binding interfaces between Pch2 and ORC present outside this fragment of Pch2 (as also suggested by our XL-MS data; see Figure 4), or alternatively could be indicative of the increased interaction of full length Pch2 with ORC due to the formation of hexamer via canonical interactions mediated by the ATPase domain of Pch2. These possibilities are indicated in the Result section: “This fragment was capable to interact with ORC (Figure 5C and D). We noted however that this interaction is much weaker as compared to the interaction with the full length Pch2 (Pch2-2-564). This could indicate additional binding interfaces between Pch2 and ORC that lie outside of this domain (as suggested by the observation of additional cross-links containing peptides from regions outside of the NTD of Pch2, and by

the residual *in vivo* interaction we observed between Pch2- Δ NTD and Orc1; see above). Alternatively, this could indicate that hexamer formation of Pch2 (driven by AAA+ to AAA+ interactions) is essential to increase the local effective concentration of the NTD)". We note that the difference in cell growth as seen in our Y2H assays should be interpreted with caution and we interpret the Y2H data only as an indication of interaction/no interaction, rather than in a quantitative manner.

6. Lines 670-672: "To test this premise, we probed the interaction between Pch2 and Orc2/Orc5 in the presence of a temperature sensitive allele of ORC1 (*orc1-161*)." For clarity, was this experiment performed at permissive or non-permissive temperature?

As indicated in the figure legends, the co-immunoprecipitation assays between Orc2-TAP and Pch2, and Orc5-TAP and Pch2-E399Q in an *orc1-161* background (Figure 6G and Supplementary Figure 8A, respectively) were performed at a permissive temperature (23°C).

References:

- Cardoso da Silva R, Villar-Fernandez MA, Vader G** (2020) Active transcription and Orc1 drive chromatin association of the AAA+ ATPase Pch2 during meiotic G2/prophase. *PLoS genetics* 16: e1008905
- Chen C, Jomaa A, Ortega J, Alani EE** (2014) Pch2 is a hexameric ring ATPase that remodels the chromosome axis protein Hop1. *Proceedings of the National Academy of Sciences of the United States of America* 111: E44-53
- Nguyen VQ, Co C, Li JJ** (2001) Cyclin-dependent kinases prevent DNA re-replication through multiple mechanisms. *Nature* 411: 1068-73
- Peng C, Luo H, Zhang X, Gao F** (2015) Recent advances in the genome-wide study of DNA replication origins in yeast. *Front Microbiol* 6: 117
- Prioleau MN, MacAlpine DM** (2016) DNA replication origins-where do we begin? *Genes & development* 30: 1683-97
- Vader G, Blitzblau HG, Tame MA, Falk JE, Curtin L, Hochwagen A** (2011) Protection of repetitive DNA borders from self-induced meiotic instability. *Nature* 477: 115-9
- Weinreich M, Liang C, Chen HH, Stillman B** (2001) Binding of cyclin-dependent kinases to ORC and Cdc6p regulates the chromosome replication cycle. *Proceedings of the National Academy of Sciences of the United States of America* 98: 11211-7
- Ye Q, Kim DH, Dereli I, Rosenberg SC, Hagemann G, Herzog F, Toth A, Cleveland DW, Corbett KD** (2017) The AAA+ ATPase TRIP13 remodels HORMA domains through N-terminal engagement and unfolding. *The EMBO journal* 36: 2419-2434
- Yeeles JT, Deegan TD, Janska A, Early A, Diffley JF** (2015) Regulated eukaryotic DNA replication origin firing with purified proteins. *Nature* 519: 431-5

August 10, 2020

RE: Life Science Alliance Manuscript #LSA-2019-00630-TR

Dr. Gerben Vader
Max Planck Institute of Molecular Physiology
Mechanistic Cell Biology
Otto-hahn-strasse 11
Max-Planck-Ring 9
Dortmund 44227
Germany

Dear Dr. Vader,

Thank you for submitting your revised manuscript entitled "Biochemical and functional characterization of a meiosis-specific Pch2/ORC AAA+ assembly". Your manuscript was re-reviewed by the original referees, and their reports are attached below. We would be happy to publish your paper in Life Science Alliance pending final revisions necessary to meet our formatting guidelines.

- please add an ORCID ID for the secondary corresponding author--you should have received instructions on how to do so
- please double-check your table legends--you have 2 legends for Table 1
- please use the [10 author names, et al.] format in your references (i.e. limit the author names to the first 10)
- please provide Figure 1E in a higher resolution
- if possible, please increase font size in Figure S7

A. FINAL FILES:

-- High-resolution figure, supplementary figure and video files uploaded as individual files: See our detailed guidelines for preparing your production-ready images, <http://www.life-science->

alliance.org/authors

B. MANUSCRIPT ORGANIZATION AND FORMATTING:

Sincerely,

Reilly Lorenz
Editorial Office Life Science Alliance
Meyerhofstr. 1
69117 Heidelberg, Germany
t +49 6221 8891 414
e contact@life-science-alliance.org
www.life-science-alliance.org

Reviewer #2 (Comments to the Authors (Required)):

The revised manuscript addresses a majority of the questions raised during the previous review. Although the inclusion of additional controls for the CX-MS experiment would still be desirable, it is acknowledged that the current COVID environment may make this difficult. No further action on the part of the authors is requested.

Reviewer #3 (Comments to the Authors (Required)):

In this manuscript, Ascensión Villar-Fernández and colleagues perform an extensive and rigorous characterization of PCH2's interaction with the ORC complex. They responded appropriately to my previous concerns in the last round of review.

I would add one suggestion: the authors hypothesize that ORC1 may play an adapter-like role for Pch2. They may want to reference van Hooff, J.J., et al., 2017 (Evolutionary dynamics of the kinetochore network in eukaryotes as revealed by comparative genomics) and Vleugel, M., et al., 2012 (Evolution and function of the mitotic checkpoint) as evidence that TRIP13's well-known adapter, p31/comet, is not conserved in yeast, to support this hypothesis.

Point-to-point reply Villar-Fernandez *et al.*

Dear Editor,

We thank the reviewers for their positive comments on our manuscript. We have addressed the small remaining editing issues, and outline detailed responses in our point-by-point reply below.

Comments from Editor:

-please add an ORCID ID for the secondary corresponding author--you should have received instructions on how to do so

We apologize for this error, the corresponding authorship of this author (EW) was erroneously added during the submission process. We have now fixed this.

-please double-check your table legends--you have 2 legends for Table 1

This has been fixed.

-please use the [10 author names, et al.] format in your references (i.e. limit the author names to the first 10)

This has been fixed.

-please provide Figure 1E in a higher resolution

We provide this now.

-if possible, please increase font size in Figure S7

We have increased the font size in this figure.

Reviewer #3 (Comments to the Authors (Required)):

In this manuscript, Ascensión Villar-Fernández and colleagues perform an extensive and rigorous characterization of PCH2's interaction with the ORC complex. They responded appropriately to my previous concerns in the last round of review.

I would add one suggestion: the authors hypothesize that ORC1 may play an adapter-like role for Pch2. They may want to reference van Hooff, J.J., et al., 2017 (Evolutionary dynamics of the kinetochore network in eukaryotes as revealed by comparative genomics) and Vleugel, M., et al., 2012 (Evolution and function of the mitotic checkpoint) as evidence that TRIP13's well-known adapter, p31/comet, is not conserved in yeast, to support this hypothesis.

We have now included these references and discuss this in the manuscript (Discussion).

August 13, 2020

RE: Life Science Alliance Manuscript #LSA-2019-00630-TRR

Dr. Gerben Vader
Max Planck Institute of Molecular Physiology
Mechanistic Cell Biology
Otto-hahn-strasse 11
Dortmund 44227
Germany

Dear Dr. Vader,

Thank you for submitting your Research Article entitled "Biochemical and functional characterization of a meiosis-specific Pch2/ORC AAA+ assembly". It is a pleasure to let you know that your manuscript is now accepted for publication in Life Science Alliance. Congratulations on this interesting work.

DISTRIBUTION OF MATERIALS:

Again, congratulations on a very nice paper. I hope you found the review process to be constructive and are pleased with how the manuscript was handled editorially. We look forward to future exciting submissions from your lab.

Sincerely,
Shachi

Shachi Bhatt
Executive Editor
Life Science Alliance
www.life-science-alliance.org